# Complex $^{40}$Ar/$^{39}$Ar age spectra from low-grade metamorphic rocks: resolving the input of detrital and metamorphic components in a case study from the Delamerian Orogen

Anthony Reid[1, 2], Marnie Forster[3], Wolfgang Preiss[1, 2], Alicia Caruso[1], Stacey Curtis[1,4], Tom Wise[1], Davood Vasegh[3], Naina Goswami[3], Gordon Lister[5]

[1]Geological Survey of South Australia, Department of State Development, GPO Box 320, Adelaide, SA 5001, Australia
[2]Department of Earth Sciences, School of Physical Sciences, University of Adelaide, SA 5005, Australia
[3]Mineral Exploration Cooperative Research Centre, Research School of Earth Sciences, The Australian National University, Canberra, ACT 2601, Australia.
[4]Mineral Exploration Cooperative Research Centre, STEM, University of South Australia, Mawson Lakes, Australia
[5]Sustainable Minerals Institute, University of Queensland, Brisbane, Australia

*Correspondence to*: Anthony Reid (anthony.reid@sa.gov.au)

**Abstract.** In this study, we provide $^{40}$Ar/$^{39}$Ar geochronology data from a suite of variably deformed rocks from a region of low-grade metamorphism within the Cambro-Ordovician Delamerian Orogen, South Australia. Low-grade metamorphic rocks such as these can contain both detrital minerals and minerals newly grown or partly recrystallised during diagenesis and metamorphism. Hence, they typically yield complex $^{40}$Ar/$^{39}$Ar age spectra that can be difficult to interpret. Therefore, we have undertaken furnace step heating $^{40}$Ar/$^{39}$Ar geochronology to obtain age spectra with many steps, so as to allow application of the method of asymptotes and limits, and recognition of the effects of mixing. The samples analysed range from siltstone and shale to phyllite and contain muscovite or phengite with minor microcline as determined by hyperspectral mineralogical characterisation. Whole rock $^{40}$Ar/$^{39}$Ar analyses were undertaken in most samples due to their very fine-grained nature. All samples are dominated by radiogenic $^{40}$Ar, and contain minimal evidence for atmospheric, Ca- or Cl-derived argon. Chloritisation may have resulted in limited recoil causing $^{39}$Ar argon loss in some samples, especially evident within the first few percent of gas release. Most of the age data, however, appear to have some geological significance. Viewed with respect to the known depositional ages of the stratigraphic units, the age spectra from this study do appear to record both detrital mineral ages and ages related to the varying influence of either cooling or deformation-induced recrystallisation. The shape of the age spectra and the degree of deformation in the phyllites suggest the younger ages may record recrystallisation of detrital minerals and/or new mica growth during deformation. Given that the younger limit of deformation recorded in the high metamorphic grade regions of the Delamerian Orogen is c. 490 Ma, the c. 470 to c. 458 Ma ages obtained in this study suggest deformation in low-grade shear zones within the Delamerian Orogen may have persisted until c. 20-32 million years after high-temperature ductile deformation in the high-grade regions of the orogen. We suggest these younger ages for deformation could reflect reactivation of older structures formed both during rift basin formation and during the main peak of the

Delamerian orogeny itself. The younger c. 470 to c. 458 Ma deformation may have been facilitated by far-field tectonic processes occurring along the eastern paleo-Pacifc margin of Gondwana.

## 1 Introduction

In many orogens, low-grade metamorphic rocks comprise the vast bulk of the surface exposure, and hence are potentially a significant source of information on the orogenic process. However, most studies into the rates and timing of orogenesis concentrate on the high metamorphic grade sections of orogens largely because high temperature minerals are either completely newly grown during orogenesis, or at least isotopically reset. Low-grade metamorphic rocks in contrast typically comprise a mixture of detrital minerals and newly grown diagenetic or metamorphic minerals (e.g. Dallmeyer et al., 1988;

Dunlap et al., 1991; Cosca et al., 1992; Dallmeyer and Takasu, 1992; Chan et al., 2000; Fergusson and Phillips, 2001; Kirschner et al., 2003; Clauer, 2013). As a result, and especially when dealing with fine-grained, low-metamorphic grade rocks, it can be difficult to separate detrital from metamorphic signals in isotopic analysis of whole rock samples.

      $^{40}$Ar/$^{39}$Ar geochronology of fine-grained and typically low-grade metamorphic rock samples poses significant methodological

challenges, including the problem of inheritance from pre-metamorphic minerals and recoil loss and $^{39}$Ar redistribution due to the fine-grained nature of the analysed minerals (Fergusson and Phillips, 2001; Phillips et al., 2012). Inheritance can be a major problem and result in over estimation of deformation ages if the age of detrital and newly grown minerals is either very close in time, or if there are limited heating steps with poor temperature control resulting in averaging of gas from different mineralogical components. One approach to minimising this problem has been to use laser ablation heating and apply this to

specific mineral grains either separated manually or analysed in situ (e.g. Chan et al., 2000; Haines et al., 2004; Kirkland et al., 2008). This can work well in the case where detrital minerals are relatively coarse grained, however, the smaller gas portions obtained via laser fusion can affect age precision.  Recoil of $^{39}$Ar during neutron irradiation can artificially reduce apparent ages and is especially problematic in fine grained samples that have undergone chloritic alteration (Lo and Onstott, 1989). Methodological attempts to minimise $^{39}$Ar recoil have been made, such as by encapsulation of samples during irradiation

(Foland et al., 1992); however, this method produces a single age per sample and is not appropriate for samples which may have multiple age populations. Therefore, resolving detrital from metamorphic signals in complex age spectra remains a challenging task, with the result that low-grade rocks are often overlooked in studies on the timing of deformation in orogenic belts.

In this paper we show that furnace step heating can yield meaningful age gradients in $^{40}$Ar/$^{39}$Ar age spectra from fine-grained whole rock samples and mica separates. In this respect, we utilise a similar methodology to Dallmeyer et al. (1988) who analysed whole rock slate and phyllite samples via furnace step heating. However, here we apply a careful heating schedule for our ultrahigh vacuum (UHV) step heating experiments, utilising a larger number of increments in the step-heating schedule

to carefully resolve any variations in age within the sample (thereby appyling methods similar to Forster and Lister, 2009; Muston et al., 2021).

Our field area, the Delamerian Orogen exposed in the Mount Lofty Ranges of South Australia, provides an ideal setting with which to investigate low-grade metamorphic rocks, because of the extensive exposure of Neoproterozoic to Cambrian metasedimentary units. The timing of orogenic activity in the Delamerian Orogen has been established based on cross-cutting relationships of igneous rocks from high-grade metamorphic zones of the orogen. The earliest interpreted syn-tectonic intrusion within the orogen has a U-Pb zircon age of $514 \pm 4$ Ma (Rathjen Gneiss; Foden et al., 1999). Bimodal intrusions that cut tectonic fabrics and are weakly or undeformed themselves were emplaced between c. 490 and 470 Ma. While the ages of these magmatic rocks are interpreted to bracket the timing of deformation between c. 514 – 490 Ma (Foden et al., 2020), there has been little geochronology from low-grade portions of the orogen to determine if deformation was synchronous across metamorphic grade zones. Our new samples yield complex age spectra that we interpret to contain evidence for ages from detrital minerals up to Mesoproterozoic in age, as well as ages from newly grown minerals of Cambro-Ordovician age related to cooling and/or deformation.

## 2 Geological setting

The Delamerian Orogen formed on the eastern proto-Pacifc margin of Gondwana in the Late Cambrian to early Ordovician and is part of the family of orogens developed during amalgamation of Gondwana (Fig. 1; Preiss, 1995a; Cawood, 2005; Foden et al., 2006; Glen and Cooper, 2021). The Delamerian Orogeny deformed Neoproterozoic to early Cambrian rocks of the Adelaide Superbasin, including the early Cambrian Kanmantoo Group and Normanville Groups (Fig. 2; Preiss, 1987; Preiss, 1995a; Flottmann et al., 1998; Preiss, 2000; Lloyd et al., 2020). Deposition of the Kanmantoo and Normanville groups was contemporaneous with the earliest arc magmatism along the Gondwanan subduction margin to the east of the Delamerian Orogen (Betts et al., 2018; Cayley and Skladzien, 2018). Deformation and metamorphism began at c. 514 Ma with emplacement of early, syn-tectonic granite (Foden et al., 1999). A belt of, high-level, undeformed granites cut major structures in the eastern parts of the orogen formed at c. 490 Ma, suggesting that the entire orogenic cycle occurred over an approximately ~24 million year period in the late Cambrian (Foden et al., 2006).

Peak metamorphic conditions of ~3-5 kbar and 550-650 °C occur in a migmatite-grade zone in the east of the orogen (Dymoke and Sandiford, 1992; Preiss, 1995a). Metamorphic isograds for the Delamerian Orogen cross some stratigraphic boundaries and folds especially within the Kanmantoo Group, supporting the notion that metamorphism was both synchronous with but also overprints some of the large-scale deformation features (Fig. 2; Offler and Fleming, 1968; Mancktelow, 1990; Dymoke and Sandiford, 1992). Folding in the orogen was also synchronous with thrusting, with thrust zones in the western region of the Delamerian displaying fault-bend folding in the hangingwall (Fig. 3; Preiss, 1995a; Flottmann and James, 1997; Flottmann

et al., 1998; Preiss, 2019). The lowest metamorphic grade regions occur along the western edge of the orogen, where weakly cleaved rocks as well as phyllitic shear zones contain metamorphic chlorite, thereby defining a chlorite-zone (Fig. 2), from where several samples analysed in this study were derived.

The few available $^{40}$Ar/$^{39}$Ar analyses from the Delamerian Orogen have mostly been interpreted to record post-peak metamorphic cooling (Turner et al., 1996; Foden et al., 2006; Turner et al., 2009; Foden et al., 2020). Laser induced step heating $^{40}$Ar/$^{39}$Ar analyses of syn-kinematic muscovite from folded schists in the Delamerian Orogen yielded a plateau age of $502 \pm 4$ Ma, interpreted to record cooling subsequent to $D_1$ thrusting (Foden et al., 2020). $^{40}$Ar/$^{39}$Ar analyses of biotite and hornblende from syn-kinematic, largely I-type granites and country rocks yield ages that range from c. 498 – 480 Ma (Turner

et al., 1996), which broadly corresponds to the timing of intrusion of late-kinematic to post-kinematic granites and mafic rocks. Turner et al. (1996) interpreted the orogen to have cooled below ~300 °C by c. 480 Ma. This is consistent with a muscovite $^{40}$Ar/$^{39}$Ar age of $478 \pm 2$ Ma from a high-level post-tectonic pegmatite (Burtt and Phillips, 2003).

In contrast to these data from the high metamorphic grade portions of the orogen, two studies have used $^{40}$Ar/$^{39}$Ar methods to

date low grade rocks from the south-western region of the orogen. Haines et al. (2004) and Turner et al. (2009) present laser fusion $^{40}$Ar/$^{39}$Ar data on detrital muscovite from the Delamerian Orogen. Their results show that many of the Neoproterozoic sedimentary rocks within the Delamerian Orogen contain detrital mica that has preserved signature of the source region and has escaped recrystallisation during orogenic activity. Turner et al. (2009) also dated white mica from a sandstone containing a differentiated cleavage and reported an $^{40}$Ar/$^{39}$Ar age of $533 \pm 15$ Ma and a Rb/Sr isochron age of $554 \pm 10$ Ma, which they

interpreted to date the timing of cleavage formation at that locality.

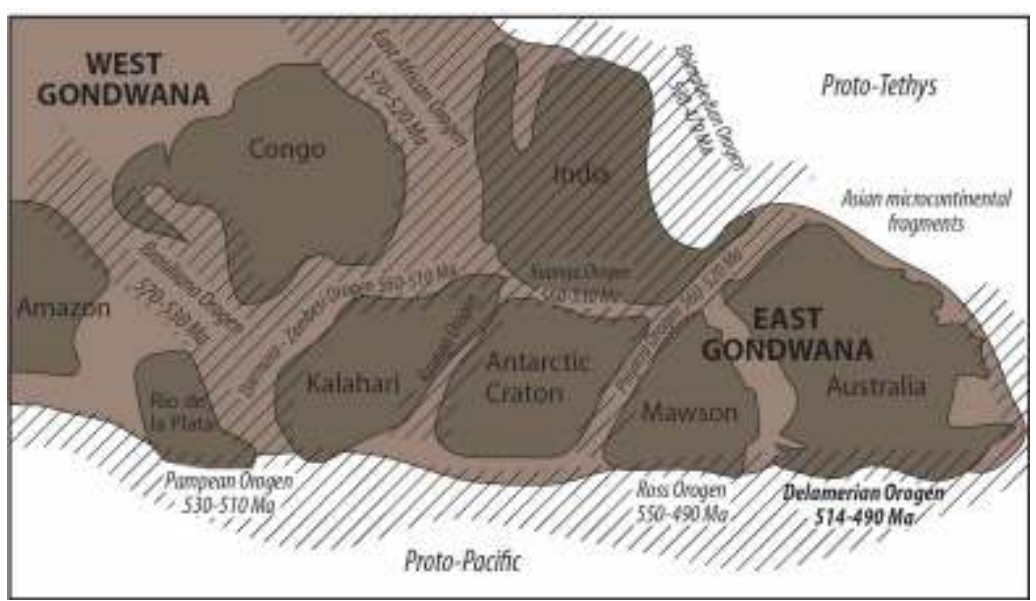

**Figure 1. Location of the Delamerian Orogen with respect to east Gondwanan terranes and Pan-African orogens. Modified from Cawood et al. (2007).**

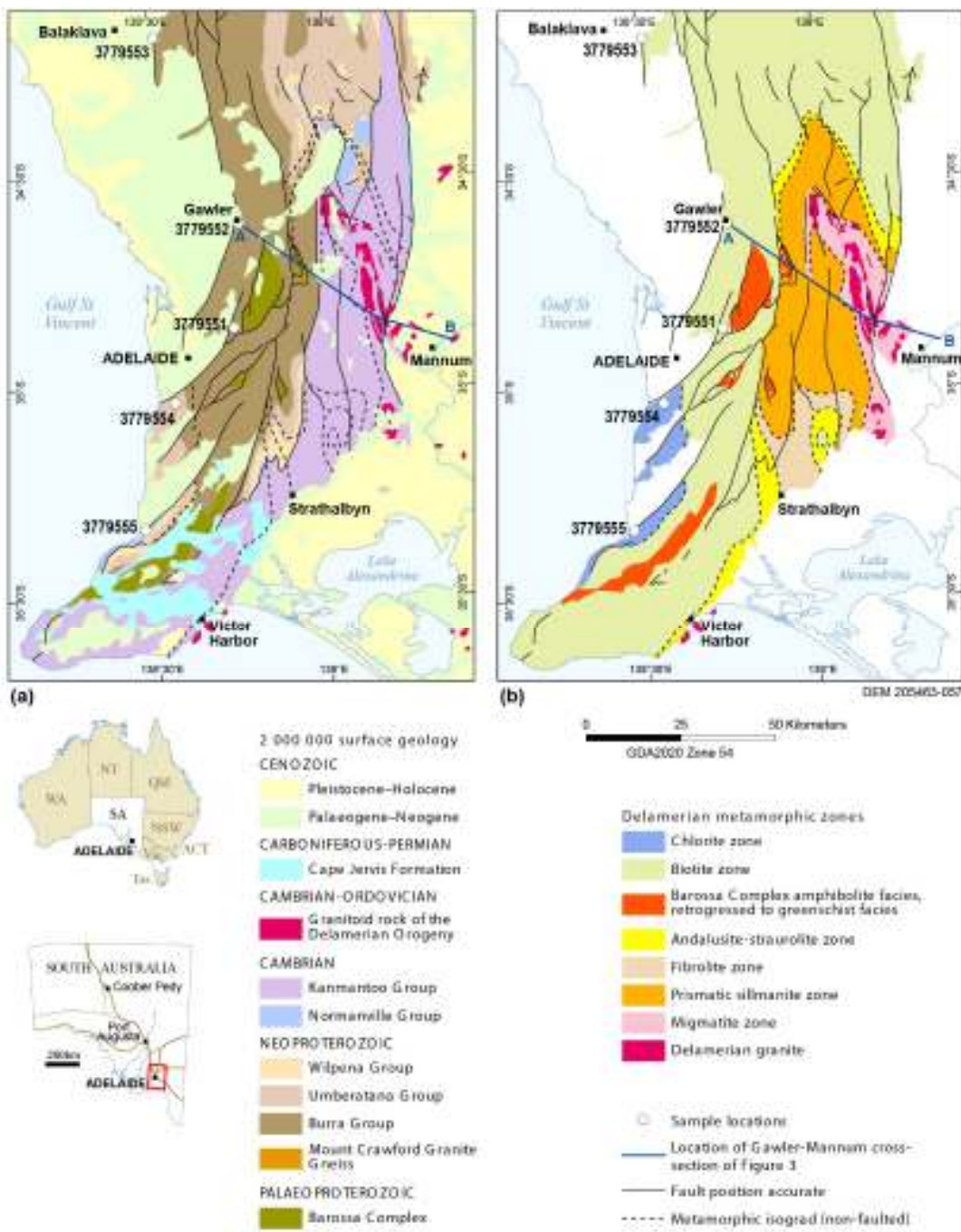

**Figure 2: Geology of the Mt Lofty Ranges, part of the Delamerian Orogen, showing location of samples analysed in this study. (a) 1:2 million scale South Australian state-wide geology dataset. (b) Delamerian metamorphic isograds of the Mt Lofty Ranges (after Offler and Fleming, 1968; Mancktelow, 1990; Dymoke and Sandiford, 1992; Preiss, 1995a).**


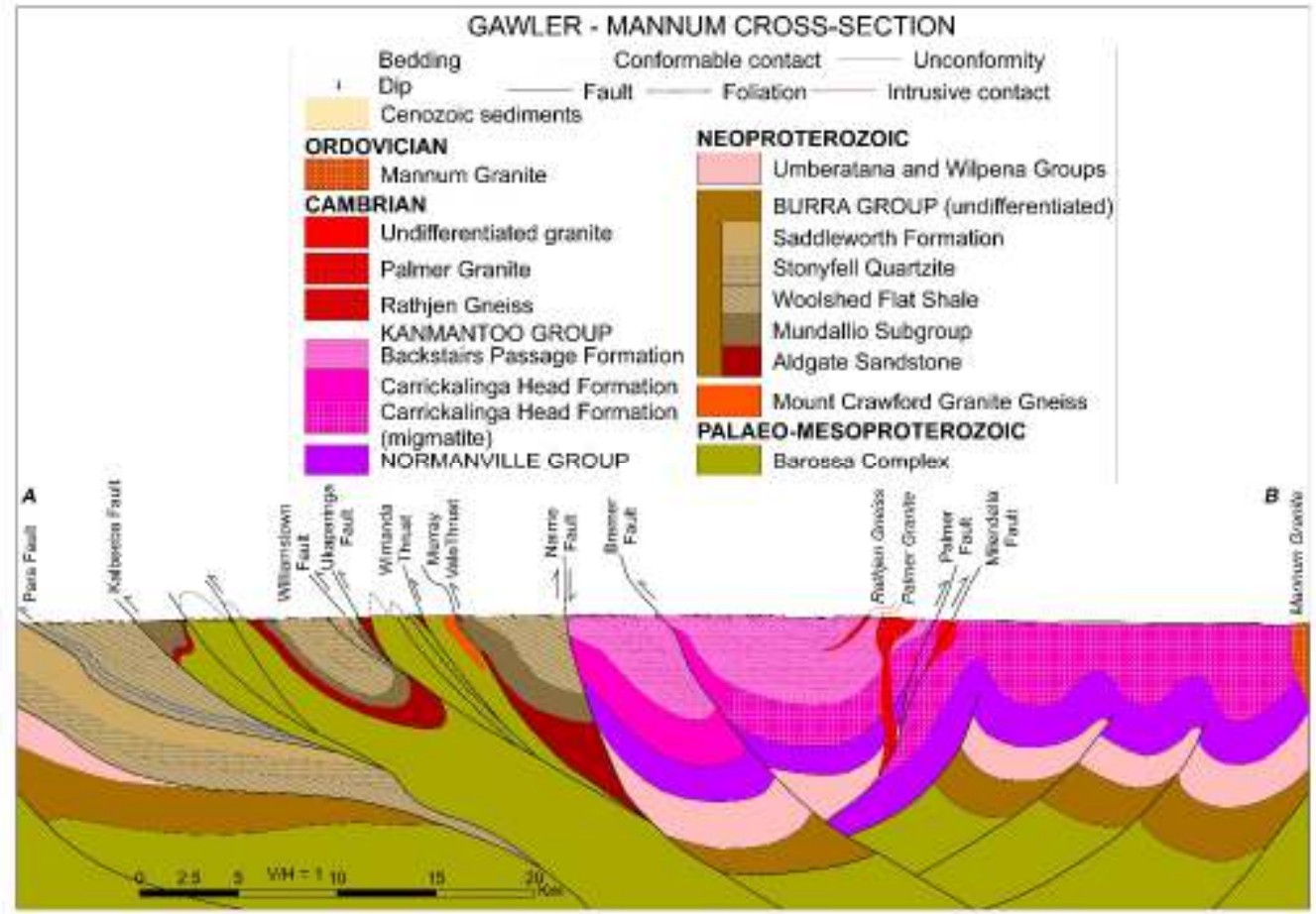

**Figure 3. Interpretative cross-section across the southern Delamerian Orogen from Gawler to Mannum. Topographic profile and surface geology, including dip of structural elements, are well constrained from previous mapping, but their extrapolation at depth in this diagram is more speculative. The left-hand portion of the cross-section illustrates the westerly vergence of the major structures**

**in the western, low-grade metamorphic zones, and the involvement of the basement. Many of the faults originated as extensional faults in the basement as a result of Neoproterozoic and Cambrian rifting. Of the mapped major faults, the Para, Kalbeeba. Williamstown, Bremer, Palmer and Milendella faults all display compressional neotectonic reactivation. The Nairne Fault differs from other faults in being a ?late Delamerian normal reactivation of a Cambrian extensional fault, and locally forms the western boundary of the Kanmantoo Trough. Metamorphic grade jumps from greenschist facies to sillimanite grade across the Williamstown**

**Fault, while migmatite is widely developed in the oldest formation of the Kanmantoo Group. Geology beneath the Kanmantoo Group is schematic only. Location of cross section shown in Figure 2, A-B.**

## 3 Sample descriptions

We have sampled rocks within the biotite and chlorite metamorphic zones of the Delamerian Orogen and analysed them using whole rock $^{40}Ar/^{39}Ar$ methods (Fig. 2). Sample details, including stratigraphic unit and location are given in Table 1. As these samples are very fine-grained we have undertaken mineralogical characterization using hyperspectral imaging with HyLogger-3™ instrumentation (Schodlok et al., 2016) at the South Australian Drill Core Reference Library, Adelaide. Full details of HyLogger-3™ methodology and mineralogical composition of samples from this study are given in Appendix A.


Samples 3779553, 3779552 and 3779551 are phyllitic schists within low angle thrust zones exposed along the western margin of the Mt Lofty Ranges (Figs 2, 4a, b, c). The samples are composed of chlorite, phengitic white mica, albite, muscovite and quartz, with minor carbonate in sample 3779551 (Appendix A). Samples 3779554 and 3779555 are siltstones from the Tapley Hill Formation and Heatherdale Shale respectively (Figs 2, 4d, e). Sample 3779554 possesses an upright, slaty cleavage and

is composed of phengitic white mica, quartz, microcline, albite, muscovite and carbonate with minor biotite and phengite (Appendix A). Sample 3779555 is very weakly cleaved and composed of phengite, kaolinite, quartz, microcline, muscovite and albite (Appendix A).

**Table 1. Details of samples analysed in this study, listed in stratigraphic order.**

| Sample | Rock type | Stratigraphic unit | Depositional age | Location | Easting | Northing | Zone (GDA94) |
|---|---|---|---|---|---|---|---|
| 3779555 | siltstone | Heatherdale Shale, Normanville Group | 514.98 ± 0.22 Ma (Betts et al., 2018) | Sellick Hill | 269649 | 6086543 | 54 |
| 3779554 | siltstone | Tapley Hill Formation, Umberatana Group | 643.0 ± 2.4 Ma (Kendall et al., 2006) | Tapley Hill | 277441 | 6120235 | 54 |
| 3779551 | phyllite | Woolshed Flat Shale, Burra Group | 790 – 730 Ma (Preiss, 2000; Lloyd et al., 2020) | Torrens Gorge | 292907 | 6140205 | 54 |
| 3779552 | phyllite | Woolshed Flat Shale, Burra Group | 790 – 730 Ma (Preiss, 2000; Lloyd et al., 2020) | Deadmans Pass | 294054 | 6167780 | 54 |
| 3779553 | phyllite | Undifferentiated Burra Group | 800 – 730 Ma (Preiss, 2000; Lloyd et al., 2020) | The Rocks | 271453 | 6216509 | 54 |


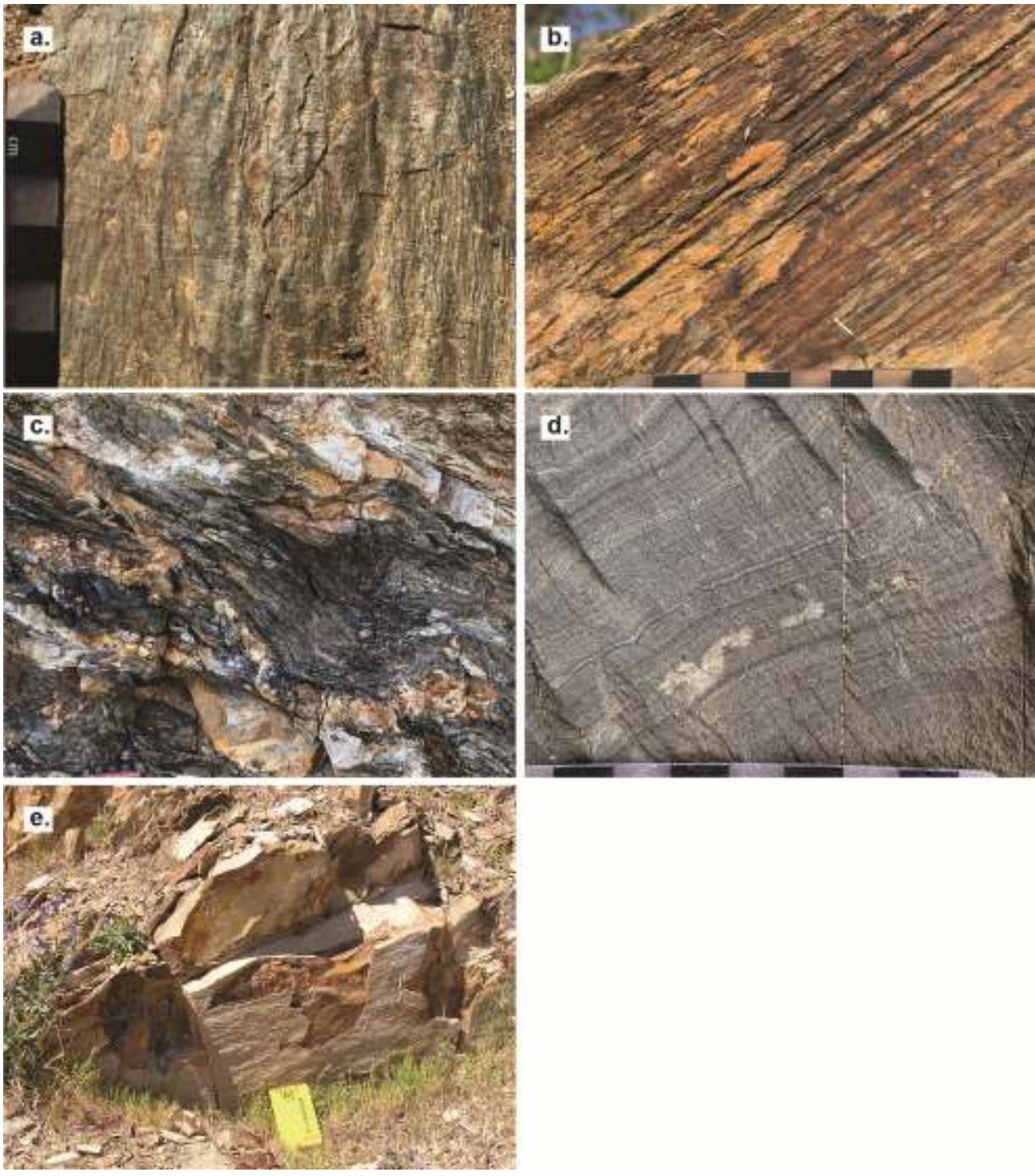

**Figure 4: Field photographs of analysed samples. a. Surface of shear fabric in phyllite from the locality of sample 3779553. b. Cross-section view of phyllite at the location of sample 3779552. View towards the south, with moderate east-dipping deformation fabric. Note orange material is lichen. c. Cross section view of phyllite of sample 3779551. This field of view shows rafts of quartzite within the phyllite and complex buckling of the phyllitic-foliation, in places with a new shear fabric developed associated with these structures. d. Detail of laminated metasiltstone of sample 3779554, showing high angle slaty cleavage developed and dipping steeply to the south-east (field of view looking towards north). e. Volcaniclastic sediment of the Heatherdale Shale from locality of sample 3779555.**

## 4 $^{40}$Ar/$^{39}$Ar analytical methods

As micas in these samples are very fine grained, whole rock samples were analysed, with the exception of sample 3779551, from which aggregates of white mica were more prominent and were able to be separated and analysed. Whole rock material was crushed and sieved to a 420-250μm size fraction. Hand picking was done to select the most suitable material/grains, before washing in de-ionised water. Necessary weights were calculated, and the weighed sample grains were packed into aluminium foils. The samples were then subjected to neutron irradiation at UC Davies nuclear reactor, USA for 12 hours and 5 minutes along with flux monitors, $K_2SO_4$ and $CaF_2$ salts for calculation of J-factors, monitoring corrections factors including $^{40}$Ar production from potassium. Biotite standard GA-1550 was used as the flux monitor. Furnace step-heating diffusion experiments were undertaken at the $^{40}$Ar/$^{39}$Ar Laboratory, Research School of Earth Sciences, Australian National University, Canberra. Complete analytical details are given in Appendix B.

Analyses were conducted using a furnace step-heating procedure through an ultrahigh-vacuum extraction line to a *Thermo Fisher* ARGUS-VI multi-collector mass spectrometer. Step-heating diffusion experiments on individual samples were carried out with a temperature-controlled furnace that allows precise control of temperature during step-heating analysis, with heating schedules between 450°C and 1450 °C. The furnace was cleaned 4 times at 1450°C for 15 minutes each time prior to each new sample being loaded ensuring no cross-contamination between samples. Then low temperature baking of each sample is done whereby the sample is heated to 420°C for 3 minutes with the gas being pumped away then heated for 12 hours at 400 °C with the gas being pumped away. This was undertaken before each analysis to remove weakly held contaminant gas from non-lattice sites. GA-1550 standards were analysed using a $CO_2$ laser and a linear best fit was then used for the calculation of the J-factor and J-factor uncertainty. Data were reduced using *Noble* 2020 software using relevant correction factors and J-factors (Appendix B). Stated precisions for $^{40}$Ar/$^{39}$Ar ages include all uncertainties in the measurement of isotope ratios and are quoted at the one sigma level and exclude errors in the age of the flux monitor GA-1550. Reported data have been corrected for system backgrounds, mass discrimination, fluence gradients and atmospheric contamination. $^{40}$K abundances and decay constants used are recommended values (Renne et al., 2011). Result tables for each step heating experiment are given in Reid and Forster (2021), with a summary of these ages and their interpretations presented in Table 2. Weighted mean ages from selected steps are calculated in the case of three or more steps forming a coherent grouping, based on Pearson's chi statistic to assess whether the scatter remains within the 95% confidence limit and calculated using the software *eArgon* (see Muston et al., 2021 for detailed description of statistical methods).

## 5 $^{40}$Ar/$^{39}$Ar age spectra results

All samples yield complex age spectra with multiple age populations (Fig. 5). Samples 3779553, 3779551a and 3779551b have similar age spectra that all have ages between c.470 and c. 459 Ma, obtained from between 20% and 70% of $^{39}$Ar released. Mean ages calculated from these successive steps produce ages of 461.2 ± 1.6 Ma, 469.9 ± 1.3 Ma, and 459.2 ± 1.1 Ma for samples 3779553, 3779551a and 3779551b respectively, each defined by 95% confidence and a Pearson's chi statistic within acceptable range for a single population (see age calculation statistics in Appendix C). In each of these samples, the age spectra then step upwards towards significantly older ages, with maxima of c. 545 Ma, c. 1172 Ma and c. 1002 Ma respectively.

Sample 3779552 yielded some of the youngest age data from the samples in this study, with ages steadily rising from as young as c. 240 Ma to an upper limit of c. 477 Ma (Fig. 5b). Finally, samples 3779554 and 3779555, from the lowest metamorphic grade portions of the orogen (chlorite zone), yield age spectra with limits of c. 508 Ma and 530.3 ± 1.4 Ma (defined by ~50% $^{39}$Ar released) respectively. These age spectra then increase in age to upper limits of c. 705 Ma and c. 567 Ma respectively. Both samples 3779554 and 3779555 have poorly defined lower limits c. 430 Ma, defined by the first 4 to 17% of $^{39}$Ar released respectively.

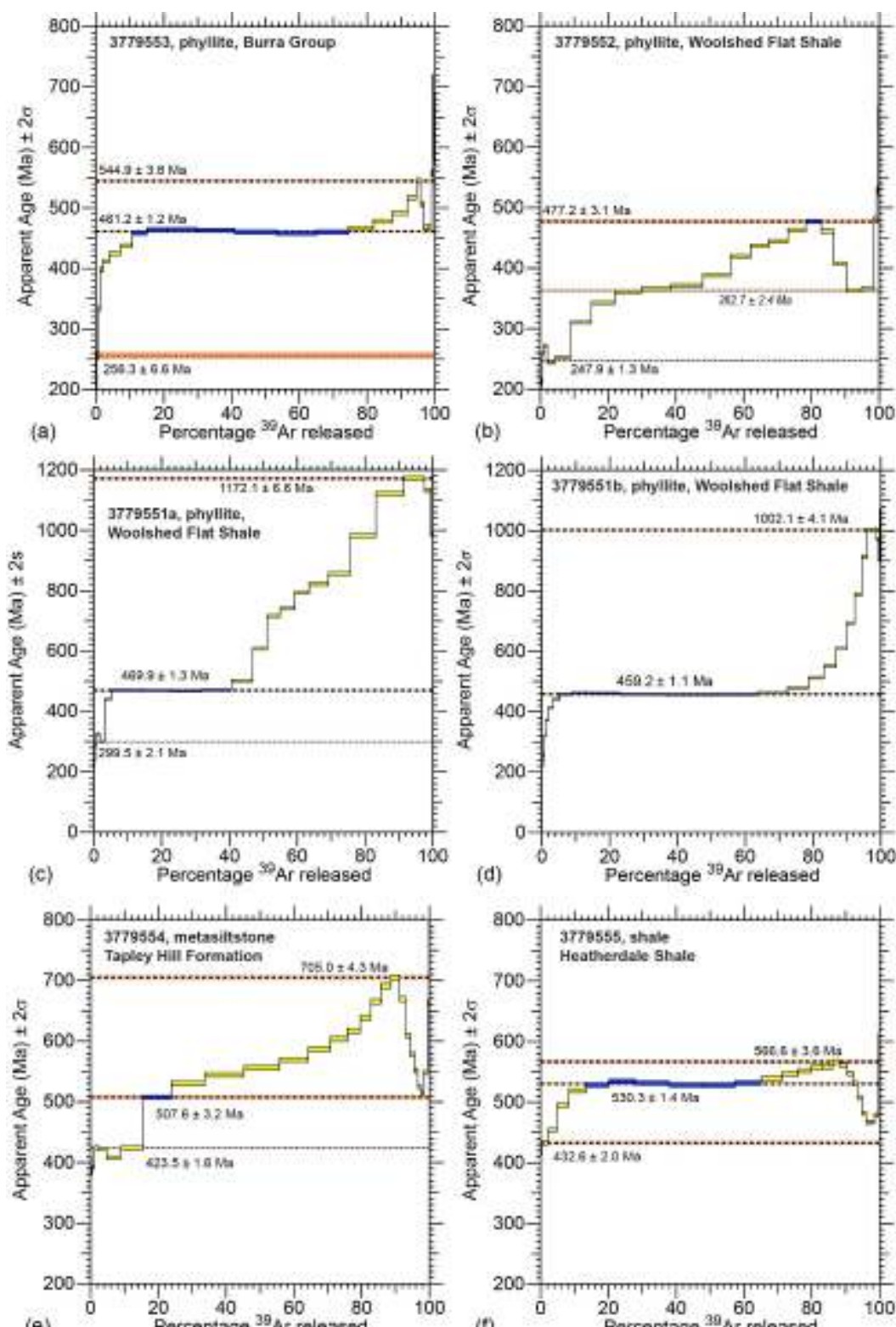


**Figure 5: Apparent age spectra for samples analysed in this study. Age limits and asymptote (italicised in sample 3779552) defined using criteria of (Forster and Lister, 2004). All data tables are given in Reid and Forster (2021) a. Sample 3779553. b. Sample 3779552. c. Sample 3779551a. d. Sample 3779551b. e. Sample 3779554. f. Sample 3779555.**

## 6 Discussion

### 6.1 Evaluation of $^{40}$Ar/$^{39}$Ar data quality from low grade metamorphic rocks

To interpret the complex $^{40}$Ar/$^{39}$Ar age spectra obtained we firstly assess the reliability of the age data with respect to isotope correlation diagrams, the relative abundance of radiogenic argon ($^{40}$Ar*) and the abundance of Ca- and Cl-derived $^{37}$Ar and $^{38}$Ar and cross-referencing this with mineralogical data derived from HyLogger$^{TM}$. Appendix C provides the complete set of relevant figures for each sample related to these isotopic data.


In each of the samples, isotope correlation diagrams between $^{36}$Ar/$^{40}$Ar and $^{39}$Ar/$^{40}$Ar show a similar pattern (Appendix C). The first heating steps begin near the $^{36}$Ar/$^{40}$Ar composition of air, track downwards towards very low values of $^{36}$Ar/$^{40}$Ar. Most samples have $^{36}$Ar/$^{40}$Ar values <0.002 for most heating steps. Significantly, the heating steps from which weighted mean ages are defined in 3779553, 3779551a, 3779551b and 3779555 all have low $^{36}$Ar/$^{40}$Ar values that cluster. Typically, the final

few steps in each sample trend towards higher $^{36}$Ar/$^{40}$Ar values, possibly indicating release of $^{36}$Ar in the final heating steps as mineral phases begin to break down under the high-temperature furnace conditions.

Mirroring the patterns in the isotope correlation diagrams, the lower temperature heating steps have correspondingly lower $^{40}$Ar*, and often elevated Ca/K and Cl/K ratios (Appendix C). In some instances, this is correlated with younger ages, which

may indicate recoil $^{39}$Ar loss affects the first few percent of the age spectra. For this reason, the ages derived from the early heating steps must be considered with caution; it is unclear if the ages from early heating steps will have geological significance. The elevated Ca and Cl values may further suggest argon from minerals such as plagioclase or chlorite may be influencing the ages from these lower temperature heating steps.

Sample 3779554 has higher Ca/K and Cl/K ratios (Appendix C) than any other sample. Interestingly, this sample has a relatively high proportion of albite compared to microcline and muscovite as recorded by thermal infrared spectroscopy (Appendix A). The HyLogger$^{TM}$ data also suggests relatively higher abundance of both calcite and dolomite in this sample, which may also be contributing to the elevated Ca/K ratios observed. The $^{40}$Ar* values are relatively consistent, and the isotope correlation diagram suggests relatively low atmospheric component, implying that most of the age data are reliable and likely

represents mixing between at least two 'virtual age components', one with an upper limit of c. 705 Ma and another with a lower limit of c. 507 Ma.

One potential issue with low grade metamorphic rocks is the presence of low temperature alteration (e.g. chlorite or clays) that can overprint the K-bearing phases such as muscovite, potentially leading to modification of the ages through recoil along fine

grained mineralogical intergrowths (e.g. Lo and Onstott, 1989; Foland et al., 1992; Di Vincenzo et al., 2003; Popov et al., 2019). Mineralogically, most samples are dominated by muscovite as the major K-bearing phase, although 3779554 and 3779555 have appreciable microcline in addition. Samples 3779551 and 3779553 contain chlorite, while sample 3779552 is dominated by chlorite (Appendix A). Ternary diagrams of $^{40}$Ar*, K-derived $^{39}$Ar and Cl-derived $^{37}$Ar, reveal that $^{40}$Ar* abundance dwarfs any contribution made by K- and Cl-derived argon, although we note the relatively large contribution of

Ca-derived Ar to sample 3779552 (Fig. 6). This, together with the discordance in the age spectrum, the variance in the isotope correlation diagram and the relatively large $^{37}$Ar contribution suggest the results from this sample need to be treated with some caution. For the remaining samples however, we can be relatively confident that the age data from the high $^{40}$Ar* heating steps is robust and can be interpreted to have geological significance.


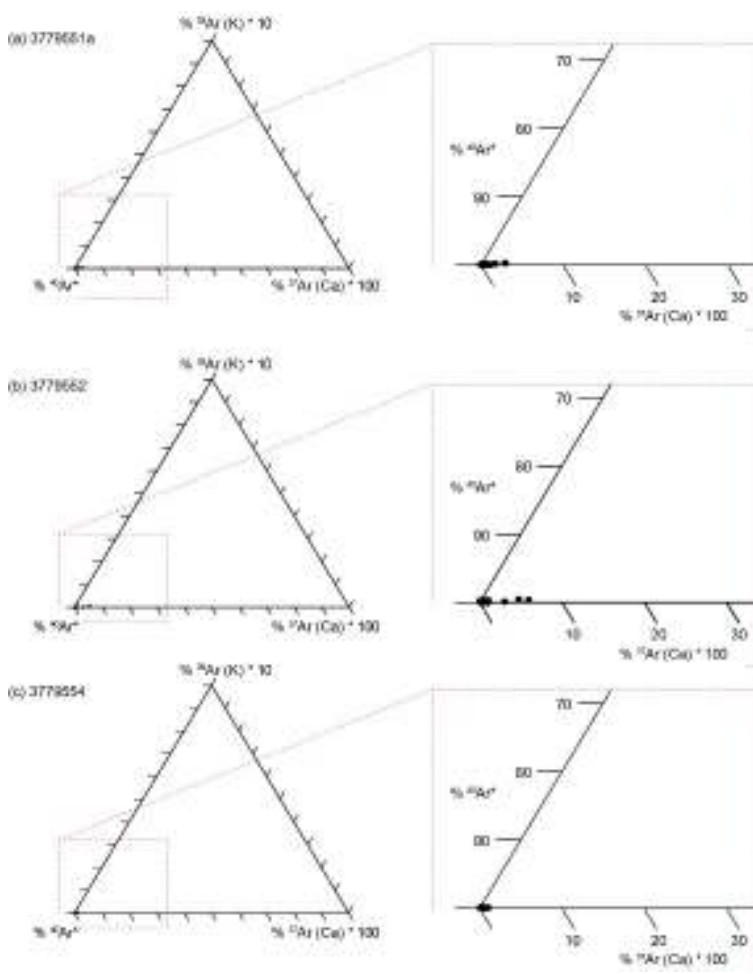

**Figure 6. Representative ternary diagram for argon step heating data for three samples from this study, (a) 3779551a, (b) 3779552 and (c) 3779554. These ternary diagrams are based on those presented by Popov et al. (2019), and plot percentages of 40Ar\*, K-derived $^{39}$Ar and Cl-derived $^{37}$Ar, where the $^{39}$Ar(K) value is multiplied by 10, and the $^{37}$Ar(Ca) value is multiplied by 100. These multiplication factors are required to make the individual heating steps visible on the ternary diagram; without these the variation in these two values is not visible, being two or more orders of magnitude smaller than $^{40}$Ar\* abundance.**

## 6.2 Interpretation of complex $^{40}$Ar/$^{39}$Ar age spectra from low grade metamorphic rocks

Now that we can say with some confidence that most of our age data is not significantly influenced by alteration, recoil, or other mineralogical effects, we turn to the problem of interpreting the complex age spectra themselves. In general, age spectra that have more than one plateau or plateau-like segment can be interpreted as containing a mixture of age populations, or virtual age components. Interpretation of such mixtures has been discussed in detail by Forster and Lister (2004), the key point being that mixtures can preserve, or partially preserve information on the timing of earlier episodes of cooling and/or deformation. Analysis of age spectra using asymptotes and limits enables maximum and minimum constraints to be placed on the timing of the various events that the rock has been subjected to (Forster and Lister, 2004).

In multiply deformed tectonites, a record of earlier microstructures can be preserved especially in the cases where the temperatures and timescale at which cooling, deformation and recrystallisation occurred are not sufficiently strong so as to completely outgas any older, relict mineral phases that may be present. Examples of this have been documented in analyses of white mica from the South Cyclades Shear Zone, Ios, where the complex age spectra reflect partial resetting during younger tectonic movement of the shear zone, in places with opposite kinematics (Fig. 7a; Forster and Lister, 2009). Similar staircase age profiles with upper and lower limits have also been measured by whole rock $^{40}$Ar/$^{39}$Ar analyses of phyllite from the Caledonian Orogen, Norway (Fig. 7b; Dallmeyer et al., 1988). In the latter example, age gradients were interpreted to reflect incomplete resetting of fine grained white mica during a Caledonian thermal event linked to emplacement of fold nappes onto the Baltic Shield (Dallmeyer et al., 1988). Numerous samples in our case study have very similar age profiles to those identified by Dallmeyer et al. (1988).

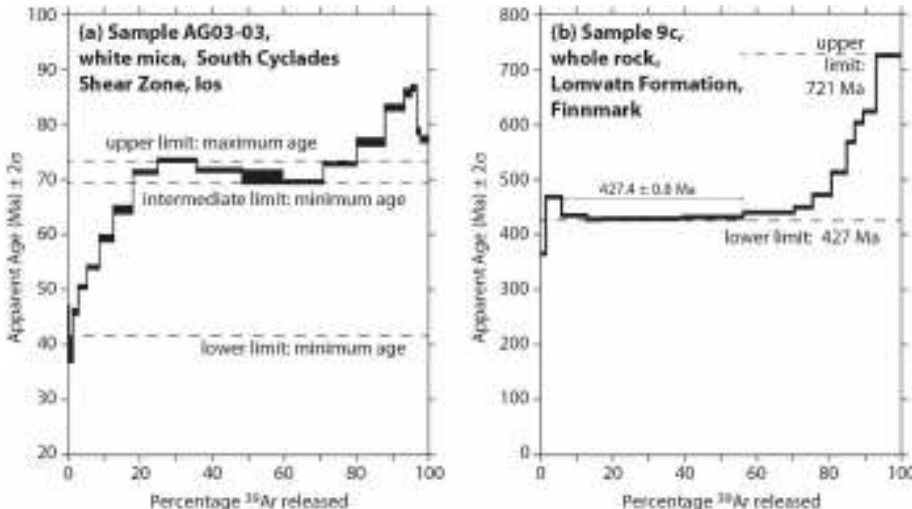

Figure 7. Example of complex age spectrum obtained from mixtures of minerals with different microstructural/microchemical compositions. (a) Sample of white mica from a South Cyclades Shear Zone, Ios, Greece, data from Forster and Lister (2009). The data are interpreted to contain three limits, and and lower limit. The upper and intermediate age limits are interpreted by (Forster and Lister, 2009) to reflect recrystallisation ages in fabrics that are relict from their earlier deformation, while the younger limit reflects partial modification of the age spectrum during Hercynian amphibolite facies metamorphism (c. 45 Ma). (b) Whole rock slate analysis from Dallmeyer et al. (1988). The sample contains both detrital mica and a cleavage with the c. 427 Ma lower limit interpreted to reflect timing of cleavage formation and the older ages the influence of detrital mineral with higher argon retentivity (Dallmeyer et al., 1988).

### 6.3 Detrital $^{40}$Ar/$^{39}$Ar signatures

Comparison of the different age spectra from our study reveals several features, which can be compared with the known depositional ages for the stratigraphic units as well as the timing of the Delamerian Orogeny established from higher grade regions of the orogen (Fig. 8). Firstly, the data reveal the presence of older ages that can be interpreted to be derived from detrital minerals preserved. The transport and deposition of K-bearing minerals can mean the $^{40}$Ar/$^{39}$Ar signature of minerals within sedimentary rocks is representative of events in the source region, rather than indicative of new mineral growth or resetting. This principle has been applied in many settings to reconstruct the early history of orogenic regions now eroded (Stuart, 2002; Blewett et al., 2019), such as in the continental foreland basin of the Himalayas (Najman et al., 1997), and the Lachlan Orogen of eastern Australia (Fergusson and Phillips, 2001).

Two samples of the Woolshed Flat Shale of the Burra Group yield upper age limits of 1002 Ma and 1172 Ma (samples 3779551a and 3779551b), which are older than the c. 790 – 730 Ma depositional age range for this unit (Preiss, 2000; Lloyd et al., 2020). Hyperspectral analysis of sample 3779551 shows the dominant K-bearing mineral in the rock is muscovite. These Mesoproterozoic ages potentially reflect input of muscovite derived from the Musgrave Province or equivalent terranes of

central Australia, as similar detrital ages are recorded in detrital zircon U-Pb studies of other rocks from the Adelaide Superbasin (Ireland et al., 1998; Keeman et al., 2020; Lloyd et al., 2020). The ages are also consistent with the range of Mesoproterozoic $^{40}Ar/^{39}Ar$ ages obtained by single grain laser fusion methods by Haines et al. (2004) from other Neoproterozoic units in the region.

A third sample from the Burra Group, also dominated by muscovite in the hyperspectral data, yields a younger upper limit of 544 Ma (sample 3779553), considerably younger than the depositional age of the unit. In this sample around 70% of the $^{39}Ar$ release produces a pooled age of c. 461 Ma, with only the final ~17% of gas release stepping upwards to older ages. This may suggest thermal and or microstructural overprinting that has had the effect of drawing down the age of pre-existing age components in a sample. Indeed, the difference in both the shape of the age spectra and the upper age limit between the two Woolshed Flat Shale samples, 3779551a and 3779551b, also suggests that thermal and or microstructural overprinting has had a variable effect on these aliquots. The difference in these age spectra suggests that variation in the degrees of rate of cooling and or degree of recrystallisation in high strain zones can affect the $^{40}Ar/^{39}Ar$ composition of fine-grained rocks, a result that mirrors the variation in recrystallisation documented in the shear zones of Ios by Forster and Lister (2009).

The remaining samples in this study yield apparent ages closer to their depositional age. Sample 3779554 yields an upper limit of 709 Ma. The maximum depositional age of the Tapley Hill Formation is 654 ± 13 Ma (U-Pb zircon; Lloyd et al., 2020) being very similar to the Re-Os date of 643.0 ± 2.4 Ma for the shale (Kendall et al., 2006), suggesting the age of the detrital minerals recorded by the $^{40}Ar/^{39}Ar$ data is at least ~55 million years older than the depositional age. This sample is a laminated metasiltstone with a spaced cleavage and contains both microcline and muscovite in similar proportions. The complexity in the age spectrum therefore relates to both variation in the relative contribution of K-feldspar and muscovite, as well as the probable influence of the formation of the spaced cleavage, which may have induced variable recrystallisation. The upper limit in the $^{40}Ar/^{39}Ar$ data is therefore a minimum estimate for the age of the detrital component.

The lowest metamorphic grade sample and the one with the least degree of strain is of the Heatherdale Shale. This unit lies at the top of the Early Cambrian Normanville Group and contains tuffaceous zircons that crystallised at 514.98 ± 0.22 Ma (Betts et al., 2018), which constrains the timing of deposition. In our data, the Heatherdale Shale sample has upper age limit of 566.6 ± 3.6 Ma, and around 50% of $^{39}Ar$ released define a single age component of 530.3 ± 1.4 Ma. Both ages are older than deposition of the Heatherdale Shale and, given the rock is weakly deformed and underwent very low-grade metamorphism (chlorite facies from the regional metamorphic grade map, see Fig. 2), these ages must represent detrital mineral ages within the sample. The age spectrum suggests that very little of the K-bearing mineral in the shale has been reset either by diagenetic modification, or by younger tectonothermal processes. The minimal modification to the apparently detrital $^{40}Ar/^{39}Ar$ ages in this sample is consistent with the Heatherdale Shale being the stratigraphically youngest and least metamorphosed rock in this sample set.

In their single grain laser fusion $^{40}$Ar/$^{39}$Ar study, Haines et al. (2004) show that the Carrackalinga Head Formation, which
directly overlies the Heatherdale Shale, has detrital muscovite ages at c. 550–600 Ma and a minor peak at c. 950 Ma. The
similarity to our data from the Heatherdale Shale supports the notion that very little resetting of the $^{40}$Ar/$^{39}$Ar system has
occurred in the detrital minerals of these units. We note, however, that assigning geological significance to the specific ages
in our data set is difficult as both muscovite and microcline is present in the sample according to HyLogger$^{TM}$ data.
Consequently, it is not certain if the ages record cooling of, for example muscovite in the source region at c. 566 Ma and
microcline at c. 530 Ma, or vice versa, or indeed if both virtual age components are mixtures of gas from both minerals. Further
work on this sample could include attempts at mineral separation at fine scale to resolve detrital $^{40}$Ar/$^{39}$Ar ages from the
Heatherdale Shale.

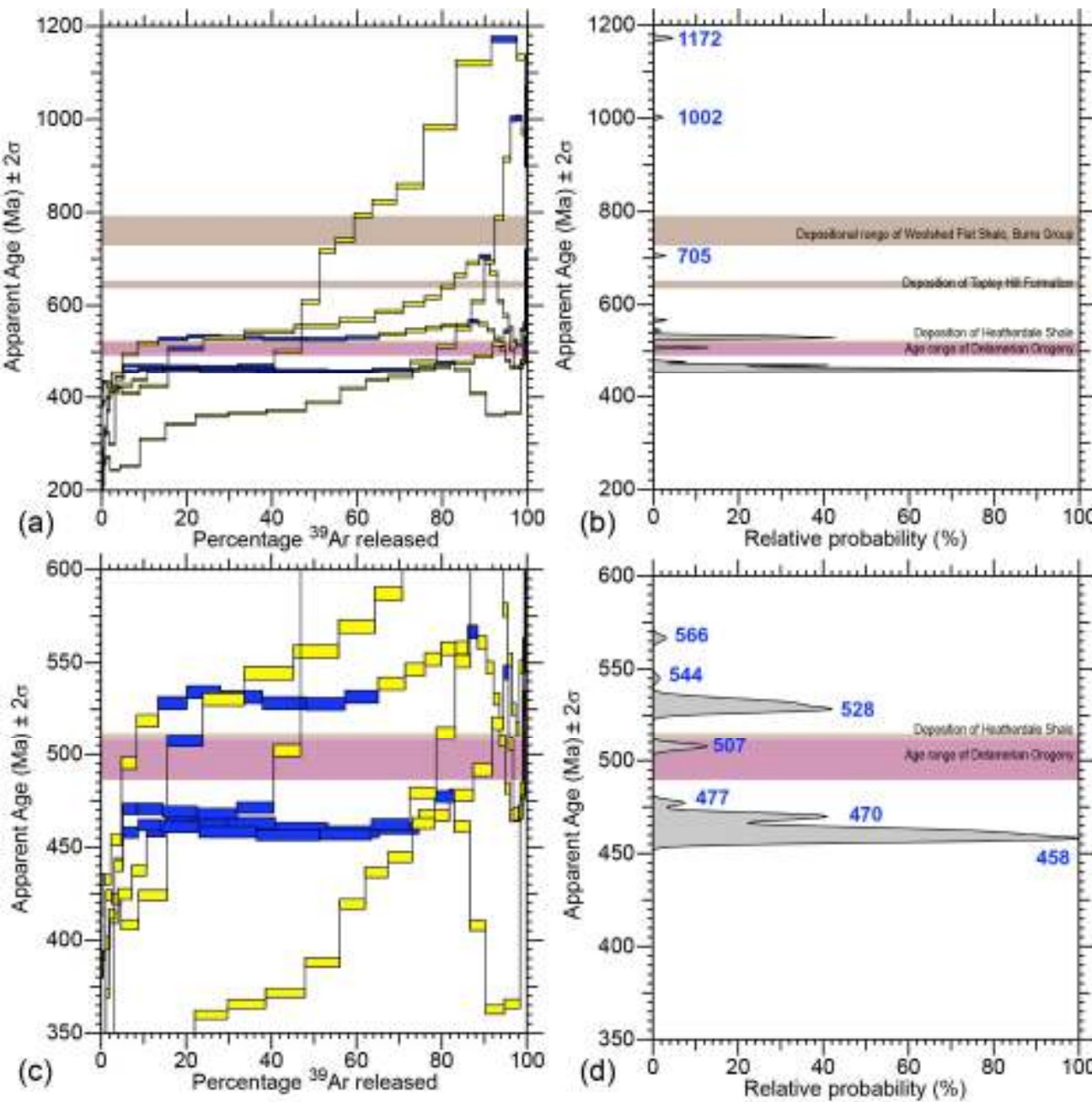

355

**Figure 8. Synoptic plots that combine all age spectra and pooled Gaussian probability plots for the limits identified in each spectrum from this study. Steps highlighted blue have been either combined into pooled ages or as age limits. Gaussian plots calculated using the program eArgon and the methods described in detail by Muston et al. (2021) (a) Synoptic plot from 200 Ma to 1200 Ma. (b)**
**Gaussian probability plot from 200 Ma to 1200 Ma. (c) Detailed synoptic plot from 350 Ma to 600 Ma. (d) Detailed gaussian probability plot from 350 Ma to 600 Ma.**

**6.4 Evidence for the c. 514 – 490 Ma Delamerian Orogeny in low grade rocks from the Delamerian Orogen**

In the Delamerian Orogen, syn-tectonic granites were emplaced at c. 514 – 500 Ma and post-tectonic granites at c. 490 - 470
Ma, which bracket the timing of deformation to between c. 514 – 490 Ma (Foden et al., 2006; Foden et al., 2020). Somewhat surprisingly, only sample 3779554 of the Tapley Hill Formation has an age limit within the c. 514 – 490 Ma time range (Fig. 7). This limit is defined by a single step at c. 507 Ma at the start of the staircase age spectrum for this sample, which then rises to older ages derived from detrital or partially reset detrital minerals as discussed above. Where we have sampled it, the Tapley Hill Formation contains a weak spaced cleavage and lies within the chlorite isograd (see Fig. 2) suggesting relatively shallow burial, and it is possible that the c. 507 Ma age in this sample reflects the time at which this spaced cleavage formed. Nevertheless, in terms of frequently measured ages from this data set, evidence for recrystallisation or cooling during the c. 514 – 490 Ma Delamerian Orogeny is conspicuously absent.

The major $^{40}Ar/^{39}Ar$ age populations from this study instead lie between c. 470 Ma and c. 458 Ma (see Fig. 7d). These derive from muscovite-and chlorite-rich phyllite samples in the north of the study area, which are more deformed than either Tapley Hill Formation or Heatherdale Shale samples. These more deformed samples all have between 35% and 55% gas release defining the ages between c. 470 Ma and c. 458 Ma. We note that sample 3779552 yields an upper limit of c. 477 Ma age, defined by a single step and is of uncertain geological significance given the discordance of the age spectrum and the abundance of chlorite in this sample. We therefore focus the following discussion on the remaining samples and the ages between c. 470 Ma and c. 458 Ma, while recognizing that further sampling may reveal age populations that are older than c. 470 Ma and therefore closer to the age of the Delamerian Orogeny recorded from higher grade zones within the orogen. The age data from this study suggest that geological process or processes were operating in the western, low-grade region of the orogen some 20 to 32 million years after deformation in the high-grade region.

While our study has shown younger ages than the main time interval indicated for the Delamerian, we note that Turner et al. (1994) and Turner et al. (2009) have also attempted to date low grade rocks from the orogen and instead produced ages older than the typically quoted ages for Delamerian orogenesis. The single muscovite laser step-heating $^{40}Ar/^{39}Ar$ age spectrum obtained by Turner et al. (2009) is very similar in form to the age spectra obtained in our study. Turner et al. (2009) averaged a series of steps early in the heating experiment to yield an age of 533 ± 15 Ma, which they interpret as the timing of cleavage formation. The higher temperature heating steps had significantly older ages (c. 850 – 900 Ma) they interpreted as detrital in origin. If we reinterpret the muscovite sample of Turner et al. (2009) using the method of asymptotes and limits (Forster and Lister, 2004), the age spectra has a lower limit of 499 ± 27 Ma (step 8 on their sample 01-HC-01). This could provide an alternative estimate of the timing of cleavage formation, with the remainder of the spectrum being a result of mixing between this younger limit and older detrital mineral ages. We note however, that Turner et al. (2009) present complimentary Rb/Sr

data on the same sample with an age of 554 ± 10 Ma, which they use to support the older age for the timing of cleavage formation in these rocks. Turner et al. (2009) rule out mixing of older and younger populations in their analysis, however considerable uncertainty remains about the significance of the Turner et al. (2009) and Turner et al. (1994) Rb/Sr isotopic data, limited as it is to a single locality (see also discussion in Preiss, 1995b).

**6.5 Cooling ages or deformation ages in fine grained samples**

Regarding the c. 470 – 458 Ma frequently measured ages obtained in our $^{40}$Ar/$^{39}$Ar data, we explore two interpretations as to their significance:

    1.   That these ages record passive cooling below the closure temperature or closure temperature range of the K-bearing minerals from an earlier deformation event that occurred above the closure temperatures of the component minerals.

2.   That these ages record the timing of recrystallisation of the minerals in the phyllite samples and therefore date the timing of deformation itself.

It is possible that the very fine-grained nature of the samples means that the K-bearing minerals in the samples may have remained open to argon diffusion longer than might be expected if the samples were coarser-grained, since the rate of diffusion is in part dependent on distance. As a result, fine-grained samples may have continued to 'leak' argon until c. 470 – 458 Ma

when they were exhumed to shallow enough crustal levels to cool sufficiently to begin to retain radiogenic argon. In such a scenario, the shape of the age spectra itself can provide information on the thermal evolution of a sample of a mixture of grains of different argon retentivity. Prolonged cooling through a range of 'closure temperatures' in the detrital minerals and newly grown metamorphic micas may be expected to yield a staircase pattern in the age spectra, as a result of different argon 'reservoirs' in the rock cooling through a range partial argon retention zones (Forster and Lister, 2004, 2010). In our study this

type of staircase pattern is best exemplified by sample 3779554, in which over 85% of the age spectrum is a mixture between a minimum of c. 507 Ma and a maximum of c. 705 Ma. The weak cleavage in this sample supports the notion that most of the K-bearing mineral in the sample is detrital, with only a minor proportion being newly grown mica, possibly grown during formation of the spaced cleavage in the Delamerian Orogeny.

If, on the other hand, more of the K-bearing minerals recrystallise during the deformation then one may expect to see a proportionally greater percentage of the age spectrum recording the younger age population. In such cases, the first portion of the age spectrum is likely to be a coherent age population (e.g. a plateau) derived from fine grained, less retentive newly grown mica, which transitions via a staircase pattern to older ages that are caused by the presence of incompletely reset/recrystallised detrital minerals. If small degrees of recrystallisation occurred, then we can reasonably expect to see only a small portion of

the gas in the age spectrum to record the recrystallisation event. In our study, samples 3779553, 3779551a and 3779551b all show staircase-like age spectra with younger plateau segments stepping upwards to older ages. These samples are the most deformed of the samples we have analysed and have the highest proportion of muscovite in the mineralogical data, with little

evidence for microcline when compared to the samples from lower strain (3779554, 3779555). One possible interpretation is that the higher degree of strain in these samples resulted in breakdown of any possible detrital K-feldspar to form newly grown muscovite (and or biotite).

In general, phyllite forms at sub-greenschist to lower-greenschist facies temperatures (~ 200–300 °C) with the dominant mineral assemblage being muscovite + chlorite + quartz (e.g. Palin and Dyck, 2021), which is the mineral assemblage of the phyllites sampled in this study. The ~ 200 – 300 °C temperature window is at the lower end of the spectrum of closure temperature windows expected for muscovite, which can be highly retentive depending on the composition (e.g. Harrison et al., 2009; Nteme et al., 2022). Therefore, we suggest that it is possible to interpret the plateau-like segments in our data from natural phyllites as recording the timing of muscovite recrystallisation and or growth in these rocks. We note also that in their study of thrust sheets in the Norwegian Caledonides, Dallmeyer et al. (1988) show age spectra with very similar profiles resulting from deformation in the footwall of a thrust, with the most deformed samples showing the increase in the percentage of the 'young' ages caused by thrusting (Fig. 6). The data presented here from the Delamerian Orogen has similar characteristics and can also be interpreted to indicate recrystallisation of mica during deformation in the zones of phyllite at c. 470 – 458 Ma. To further investigate the possibility that the ages recorded in our study from the zones of phyllite are a result deformation, more samples could be taken progressively further away from the thrust zones to investigate changes in argon age related to variation in factors such as deformation intensity, grain size and mineralogy of the shear zones.

If we interpret at least three of the age spectra presented here as recording the timing of deformation in the phyllitic thrusts at the western edge of the Delamerian Orogen, this implies that deformation in these thrust sheets was active at c. 470 – 458 Ma, which is at least 20 million years after the cessation of deformation in the high metamorphic grade regions of the orogen recorded in the ages of syn- to post-tectonic igneous intrusion (Foden et al., 2006). This interpretation produces several complications regarding how to account for the structural geology of the region. In the western Delamerian Orogen, cleavage foliations are axial planar to major regional folds with the intensity of cleavage development increasing and the orientation of the cleavage becoming shallower (east-dipping) especially on lower fold limbs, culminating in shear zones such as the ones sampled in this study (see fault-related folding in cross section of Figure 3 and also balanced cross sections of Flottmann and James, 1997; Flottmann et al., 1998). While it is possible that out of sequence thrusting could have occurred late in the deformation history after the main period of Delamerian deformation, the continuity of structure from low to high metamorphic grade regions across the orogen, raises the possibility that the c. 470 – 458 Ma ages record only the last phase of deformation in these thrust sheets, rather than the timing of inception of the thrusts themselves. We suggest a likely scenario is that these deformation zones are reactivated zones of weakness formed during earlier phases of the Delamerian Orogeny and possibly even earlier, as some of these structures formed during the Neoproterozoic and Cambrian rifting (Preiss, 1995a; Flottmann and James, 1997; Flottmann et al., 1998; Preiss, 2019). Therefore, although the c. 470 – 458 Ma ages are younger than the main ages for Delamerian deformation elsewhere in the orogen and were a surprising outcome of this study, there is no reason why

these fault zones could not have been active, or continued to be active, during younger periods of deformation. Indeed, continued reactivation of older structures throughout the Phanerozoic is a feature of faults of the Mt Lofty Ranges, as detailed by Preiss (2019).


The c. 470 – 458 Ma activity recorded in our samples from thrusts of the western portion of the Delamerian Orogen is potentially related to far-field tectonic drivers along the then active eastern margin of Gondwana, now preserved in the adjacent Lachlan Orogen. Whilst tectonism associated with plate-margin processes to the east prior to c. 470 Ma had been primarily focused into the locally weak continental rift and back-arc basin, stabilisation of the Delamerian Orogen by c. 480 Ma may

have facilitated far-field stresses being then transmitted across the whole orogen from the paleo-Pacific eastern margin of the Australian continent. The paleo-Pacific underwent several cycles of arc magmatism, accretionary tectonics, deformation and slab rollback during the Paleozoic (e.g. Glen, 2013; Rosenbaum, 2018, and others), and is a type example of an extensional accretionary orogen (Collins, 2002; Kemp et al., 2009). Between c. 511 Ma to c. 500 Ma arc magmatism was concentrated in the Staveley Arc adjacent a west-dipping subduction system (Cayley and Skladzien, 2018; Lewis et al., 2018) and was broadly

contemporaneous with the Delamerian Orogeny in the South Australian portion of the system. Intra-oceanic arc magmatism in the Macquarie Arc initiated outboard of this system as early as c. 503 Ma and continued to evolved through the Ordovician and into the Silurian (Kemp et al., 2020). Intriguingly, detailed stratigraphy and geochronology of the Macquarie Arc shows a series of 'phases' of magmatism and sedimentation, with the initial phase followed by a volcanic hiatus of c. 8 Myr, between c. 474 – 466 Ma (Glen et al., 2012). Although no precise correlation is necessarily implied, this hiatus in the Macquarie Arc

partly corresponds to the timing of deformation inferred from the thrust zones in the Delamerian Orogen dated in this study. This may suggest there was some type of mechanical link between subduction dynamics along the paleo-Pacific margin and the activity along Delamerian thrust systems far inboard from the active margin.

## 7 Conclusion

Isotopic analyses of low-grade metamorphic rocks can be difficult to interpret as these are a mixture of detrital and newly

grown, diagenetic or metamorphic minerals. As a consequence of this mixing, the isotopic signals from each of these components are not necessarily easy to resolve. In this study, we have applied the $^{40}$Ar/$^{39}$Ar method with careful furnace step heating analysis of whole rock samples of low-grade metamorphic rocks. Key to interpreting the complex $^{40}$Ar/$^{39}$Ar age spectra thus derived is the idea that the shape of the spectra themselves is informative as to the nature of the various gas reservoirs in a sample, and limits within the spectra define minimum and maximum ages for these components (Forster and Lister, 2004;

Forster and Lister, 2009). In addition, examination of isotope correlation diagrams and ternary plots of $^{40}$Ar*, $^{39}$Ar(K) and $^{37}$Ar(Ca) reveal the majority of gas in our samples is indeed radiogenic $^{40}$Ar*, with the possible exception of one sample that has a significant proportion of chlorite. Data from this study shows that the Neoproterozoic rocks of the southern Delamerian

Orogen preserve detrital mineral components up to c. 1172 Ma in age, while Cambrian shale yields detrital mineral components with ages of c. 567 and c. 530 Ma, closer to the age of deposition.


The younger limits in our data occur in phyllitic samples with age segments at c. 470 – 458 Ma, which is at least 20 million years younger than the timing of the Delamerian Orogeny documented from studies of granitic intrusions and amphibolite-grade metamorphic fabrics (Foden et al., 2006; Foden et al., 2020). Two interpretations of the younger age components are (1) that they record regional cooling, or (2) record of deformation-induced recrystallisation at temperatures cooler than the closure

temperature windows for the main K-bearing minerals in the phyllite. The second interpretation implies that some zones of phyllite in the Delamerian Orogen were active after the main phase of deformation recorded in higher metamorphic grade regions of the orogen. Potential coupling of stress transmitted from the distal plate margin along the paleo-Pacific margin of eastern Gondwana with the far-field continental interior may explain the (re)activation of some shear zones in the Delamerian Orogen. Further work is required to test these hypotheses, however, the new data point to the possibility that widespread

application of this methodology, likely in combination with other methods such as Rb-Sr dating (e.g. Zack and Hogmalm, 2016), is a pathway forward for analysis of low-grade metamorphic rocks. Certainly, the $^{40}Ar/^{39}Ar$ method is complementary to analysis of metamorphic rocks form higher grade regions within orogens by other isotopic methods (e.g. zircon, monazite U-Pb) and provides for an holistic understanding of orogenic regions.

**Appendix A. Hyperspectral mineralogical methods and results**

*Hyperspectral mineralogy methods*

Samples were mineralogically characterised with a HyLogger-3™ (Schodlok et al., 2016), instrument at the Geological Survey of South Australia, Drill Core Reference Library, Adelaide. The HyLogger-3™ collects multiple data types including reflectance from the Visible-Near Infrared, Shortwave Infrared (VNIR-SWIR, 380-2,500 nm) and Thermal Infrared (TIR,

6,500 - 14,000 nm), high-resolution imagery and laser profilometer data at 4 mm resolution. Although this instrument is primarily designed to scan drill core, it is adaptable to collect data from a variety of geological materials. Each rock sample was measured with five passes of the HyLogger™. The resulting spectral data was processed to exclude noisy and unwanted spectra using The Spectral Geologist™ v8.1.0.3 (TSG) software (Mason et al., 2020). Two algorithms, The Spectral Assistant™ (TSA) and joint Constrained Least Squares (jCLST), unmix the of spectral data and produces a semi-quantitative

mineralogical result for SWIR and TIR. In this study, spectral results were obtained via traverses across each sample, with an average mineralogical composition compiled from both the SWIR and TIR spectrometers, as each wavelength range is diagnostic for different minerals. Note that different minerals can appear in both SWIR and TIR analysis and that the total mineralogy identified by each spectral range is not necessarily indicative of the entire mineralogy of the sample. These data

are used as a guide to the major mineral components, in particular the fine-grained minerals such as mica that are significant
for the present $^{40}$Ar/$^{39}$Ar study.

Scalars are algorithms applied within TSG designed to interpret diagnostic absorption features related to specific mineral species. A scalar developed by Haest et al. (2012) examines the wavelength change of the 2,200 nm Al-OH absorption feature (Figure A1), known to distinguish endmembers of white mica. Shorter wavelengths are indicative of a muscovitic composition and longer wavelengths demonstrate a phengitic white mica.

HyLogger$^{TM}$ results for each sample are given in the following and are shown as:
  a) plain light high-resolution image of each sample,
  b) pie diagram compositional summary representing semi-quantitative HyLogger™ data identified in the SWIR wavelength range,
  c) pie diagram compositional summary representing semi-quantitative HyLogger™ data identified in the TIR wavelength range.

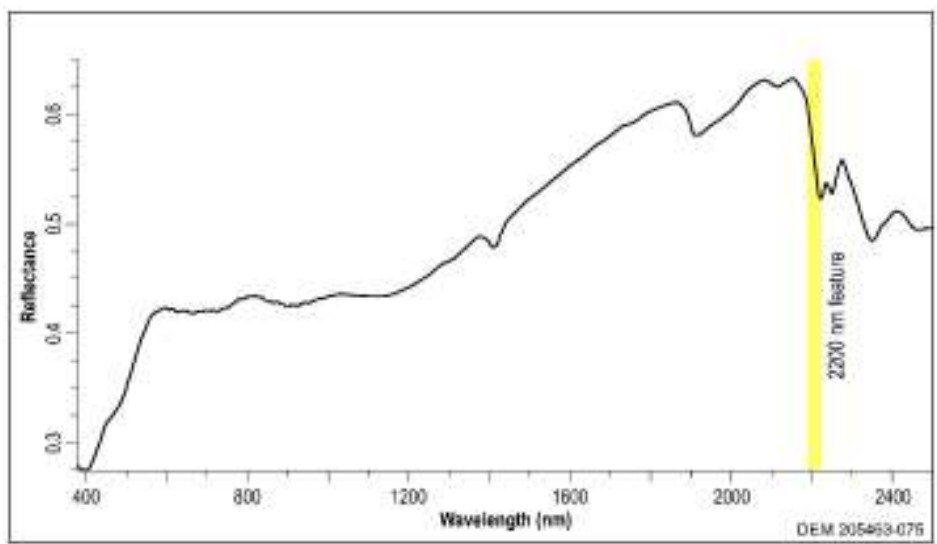

**Figure A1. A representative spectral measurement from a sample in this study, the 2,200 nm feature is highlighted to demonstrate its position relative to the VNIR-SWIR spectral signature.**

*Hyperspectral mineralogy results for each sample analysed in this study*

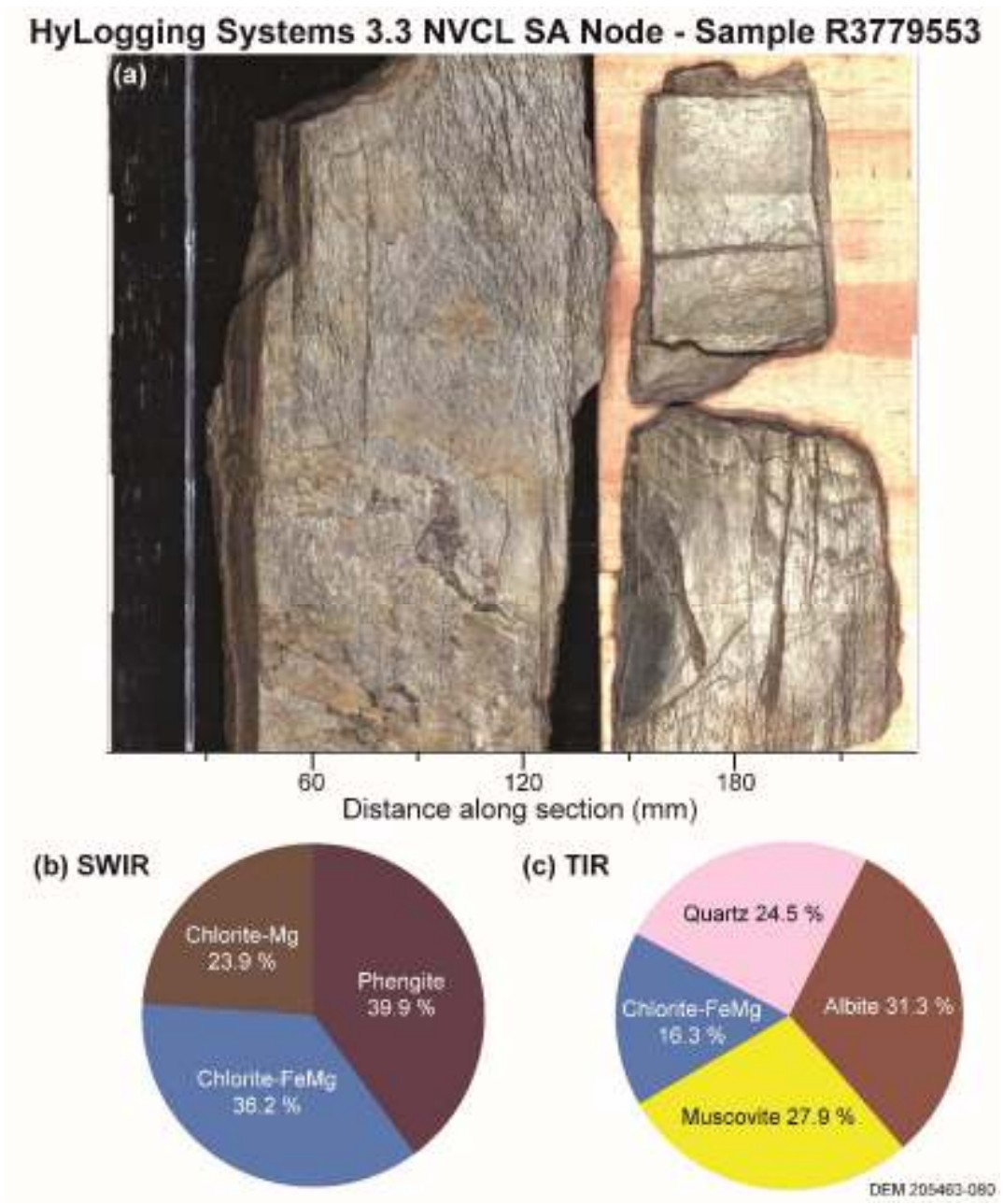

**Figure A2. Hyperspectral mineralogy results for sample 3779553.**

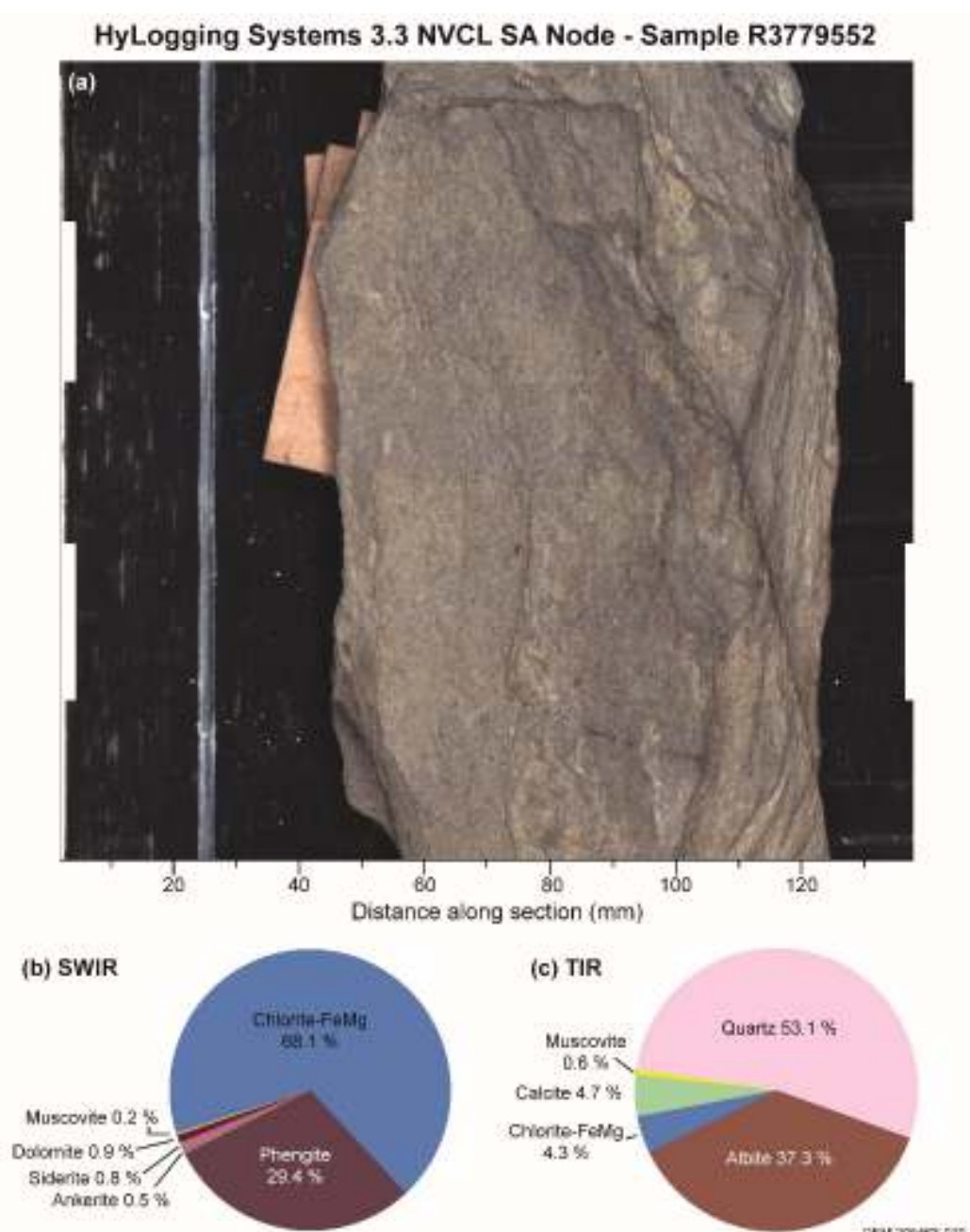

Figure A3. Hyperspectral mineralogy results for sample 3779552.

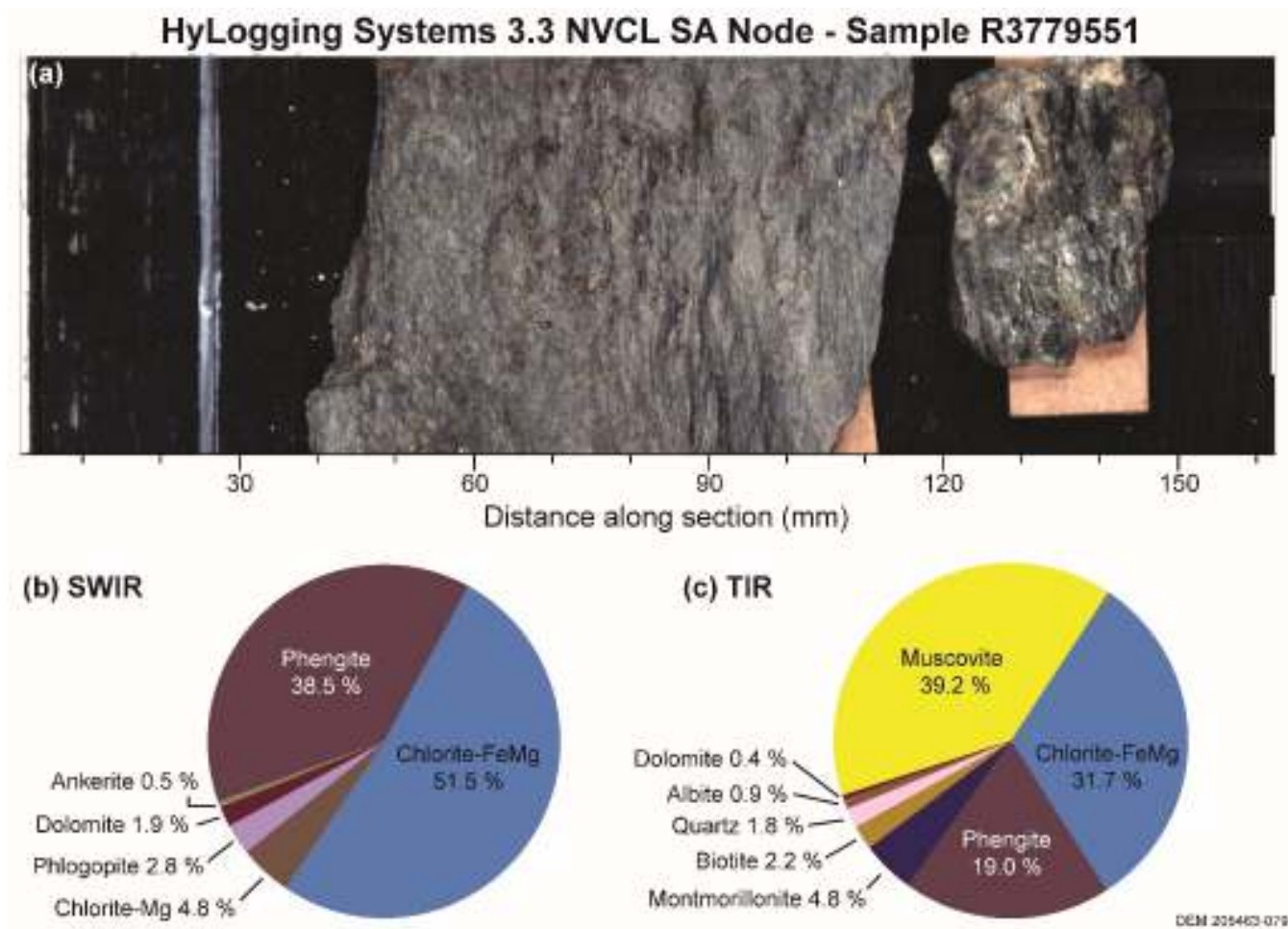

**Figure A4. Hyperspectral mineralogy results for sample 3779551.**

**Figure A5. Hyperspectral mineralogy results for sample 3779554.**

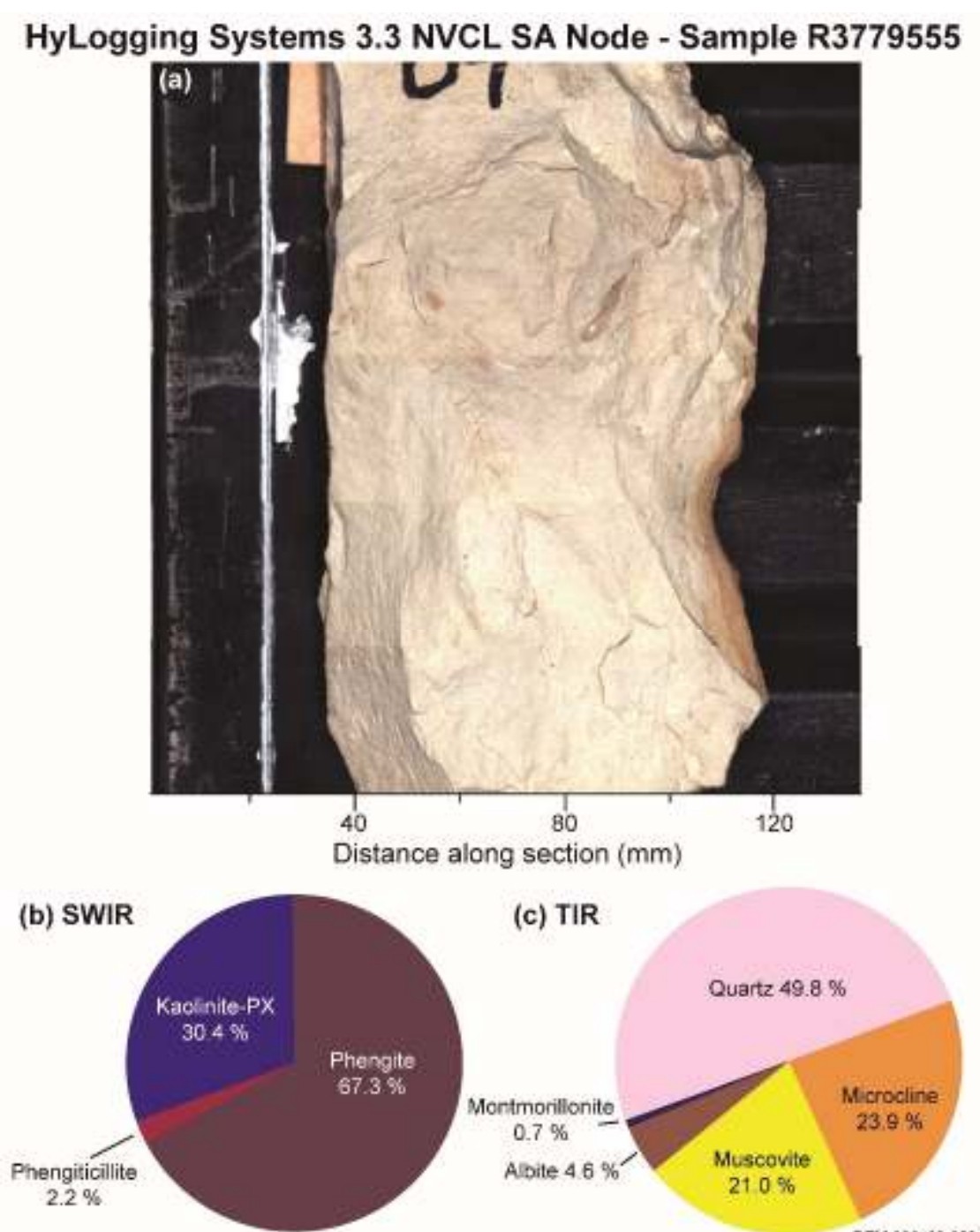

**Figure A6. Hyperspectral mineralogy results for sample 3779555.**

**References for Appendix A:**

Haest, M., Cudahy, T., Laukamp, C., and Gregory, S.: Quantitative Mineralogy from Infrared Spectroscopic Data. I. Validation of Mineral Abundance and Composition Scripts at the Rocklea Channel Iron Deposit in Western Australia, Economic Geology, 107, 209-228, 10.2113/econgeo.107.2.209, 2012.

Mason, P., Berman, M., Guo, Y., Warren, P., Lagerstrom, R., Bischof, L., Huntington, J., and Rodger, A.: The Spectral Geologist (8.1.0.3), CSIRO [code], 2020.

Schodlok, M. C., Whitbourn, L., Huntington, J., Mason, P., Green, A., Berman, M., Coward, D., Connor, P., Wright, W., Jolivet, M., and Martinez, R.: HyLogger-3, a visible to shortwave and thermal infrared reflectance spectrometer system for drill core logging: functional description, Australian Journal of Earth Sciences, 63, 929-940, 10.1080/08120099.2016.1231133, 2016.

## Appendix B. Details of $^{40}Ar/^{39}Ar$ analytical methods

**Sample irradiation details:**

Irradiation of samples for $^{40}Ar/^{39}Ar$ analysis is required, this was undertaken at the University of California Davis McClellan Nuclear Research Centre, CA, US. The samples in this study were wrapped in aluminium foil and irradiated in ANU CAN #36 and the canister had 1.0mm cadmium shielding (Tetley et al., 1980). For each sample, calculated amounts of grains were weighed, recorded and wrapped in labelled aluminium packets in preparation for irradiation. The sample filled foils were placed into a quartz irradiation canister together with aliquots of the flux monitor GA1550. The GA1550 standards are dispersed throughout the irradiated canister, between the unknown age samples. In addition, packets containing $K_2SO_4$ and $CaF_2$ were placed in the middle of the canister to monitor $^{40}Ar$ production from potassium. Irradiated samples were unwrapped upon their return to the Australian National University, and then rewrapped in tin foil in preparation for analysis in the mass spectrometer.

**$^{40}Ar/^{39}Ar$ procedures and analysis information:**

Samples and standards were analysed in the Argon Laboratory at the Research School of Earth Science, The Australian National University, Canberra, Australia using a *Thermo Fisher* ARGUS-VI multi-collector mass spectrometer. A furnace step-heating technique was used to extract argon isotopes from the samples to ensure 100% release of $^{39}Ar$, while the flux monitors crystals (GA1550 biotite) were fused using a $CO_2$ continuous-wave laser; gases extracted from both the samples and standards were analysed in the Argus VI mass spectrometer. For the furnace step-heating process, the samples were wrapped in tin foil so as to melt the tin and pump away the gases prior to the sample analysis. The furnace was degassed 4 times at 1,450°C for 15 minutes and the gas pumped away prior to the loading of the subsequent sample. Gas released from flux

monitors and each step of sample analyses was exposed to three different Zr-Al getters to remove active gases for 10 minutes and the purified gas was isotopically analysed in the mass spectrometer. Samples were analysed with 30 steps and with temperatures of the overall schedule rising from 450° to 1,450°C (Reid and Forster, 2021). The $^{40}Ar/^{39}Ar$ dating technique is adapted from Mcdougall and Harrison (1999) and described in Forster and Lister (2009).

The background levels were measured and subtracted from all analysis, laser and furnace. For example, backgrounds were measured prior to every step of the sample analysis and subtracted from the isotope intensities for $^{40}Ar$, $^{39}Ar$, $^{38}Ar$, $^{37}Ar$ and $^{36}Ar$. The nuclear interfering values for the correction factors for the isotopes are listed below. These are measured for the reactions and uncertainties of $(^{36}Ar/^{37}Ar)_{Ca}$, $(^{39}Ar/^{37}Ar)_{Ca}$, $(^{40}Ar/^{39}Ar)_K$, $(^{38}Ar/^{39}Ar)_K$ and $(^{38}Ar)_{Cl}/(^{39}Ar)_K$, and were calculated prior to sample analysis. The decay constant (Lambda$^{40}K$ = 5.5305E-10) is based on the reported values in Renne et al. (2011).

| ANU IRRADIATION CAN #36 | |
|---|---|
| Flux Monitor:  GA1550 @ 99.18 ± 0.142 Ma (Intercalibration from Spell & McDougall 2003) | |
| $(^{36}Ar/^{37}Ar)_{Ca}$ correction factor | 1.012835E-4 |
| $(^{39}Ar/^{37}Ar)_{Ca}$ correction factor | 8.469432E-4 |
| $(^{40}Ar/^{39}Ar)_K$  correction factor | 1.340092E-1 |
| $^{38}Ar/^{39}Ar)_K$  correction factor | 1.054454E-2 |
| $(^{38}Ar)_{Cl}/(^{39}Ar)_K$ correction factor | 8.184720E-2 |
| Ca/K conversion factor | 1.90 |
| Discrimination factor | 1.00441 ± 0.185% |
| Lambda $^{40}K$ | 5.5305E-10 |
| Total irradiation power | 12.08 MW |
| Irradiation Date | August 11-12, 2020 |

| Sample | Field Way Point | Foil | J-Factor | J-Factor uncertainty | Mineral | Measurement Date |
|---|---|---|---|---|---|---|
| 3779554 | WP 12-01 | M03 | 2.24374E-03 | 0.2425 | Whole Rock | 22 Mar,2021 |
| 3779555 | WP 13-01 | M11 | 2.23290E-03 | 0.2425 | Whole Rock | 24 Mar,2021 |
| 3779552 | WPT 09-01 | M17 | 2.22762E-03 | 0.2426 | Whole Rock | 30 Mar,2021 |
| 3779553 | WPT 10 | M21 | 2.22250E-03 | 0.2426 | White Mica | 12-Mar-2021 |
| 3779551a | WPT 5-01 | M27 | 2.21396E-03 | 0.2427 | Whole Rock | 05 Apr,2021 |
| 3779551b | WPT 5-02 | M28 | 2.21225E-03 | 0.2427 | Whole Rock | 06 Apr,2021 |

$^{40}K$ abundances and decay constants are calculated per Renne et al. (2011). Stated precisions for $^{40}Ar/^{39}Ar$ ages include all uncertainties in the measurement of isotope ratios and are quoted at the one sigma level and exclude errors in the age of the

fluence monitor GA1550. The age of the flux monitor (GA1550 = 99.18 Ma) is based on the intercalibration value between GA1550 and Fish Canyon Sanidine age reported in Spell and Mcdougall (2003). The reported data have been corrected for system backgrounds, mass discrimination, fluence gradients and atmospheric contamination. GA1550 standards were analysed and a linear best fit was then used for the calculation of the J-factor and J-factor uncertainty.

Data reductions were done with an adapted version of *Noble* Software (2020, written and adapted by the Australian National University Argon Laboratory). The data reduction was based on optimising MSWD (the mean square of weighted deviates) of isotopes intensities with an exponential best fit methodology. The discrimination factor was calculated by analysing five Air Shots analysis on either side of sample analysis, based on the atmospheric $^{40}Ar/^{36}Ar$ ratio (298.57; see Lee et al., 2006), and the calculation of the 1amu was used for the discrimination factor.

$^{40}Ar/^{39}Ar$ isotopic data of the sample is supplied in the Reid and Forster (2021), which includes details on: the heating schedule, Argon isotopes abundances and uncertainty levels, %Ar*, $^{40}Ar*/^{39}Ar(K)$, Cumulative $^{39}Ar$%, Age and uncertainty, Ca/K, Cl/K, J-factor and J-factor uncertainty, noting that the fractional uncertainties are shown as %, and are stated in the headings of the appropriate columns. Uncertainty levels of the calculated ages are at one sigma. Components involved in the calculation of the uncertainties are listed in the table below.

| Uncertainty | Components involved in the calculation |
|---|---|
| Isotope Abundances | Uncertainty of isotope measurement<br>Uncertainty of Mass Discrimination Factor (except for $^{39}Ar$) |
| J-Factor | Uncertainty of $^{40}K$ Decay Constant<br>Uncertainty of Age of the Flux monitor<br>Uncertainty of Flux monitor isotopes abundances |
| Calculated Age | Uncertainty of Isotopes Abundances<br>J-Factor value and uncertainty of J-Factor<br>$^{40}K$ Decay Constant value and uncertainty of $^{40}K$ Decay Constant |

**Appendix C. Isotopic data plots for samples of this study.**

For each sample, the age spectrum (a), isotope correlation diagram (b), percentage radiogenic $^{40}$Ar ($^{40}$Ar*; c), Ca/K ratio (d) and Cl/K ratio (e) are presented. Below each set of plots is a description of the ages selected for combination or as single steps, along with statistical information on the pooled ages and the steps aggregated. Note that limits and asymptotes defined after Forster and Lister (2004); age plateau as per Schaen et al. (2020).

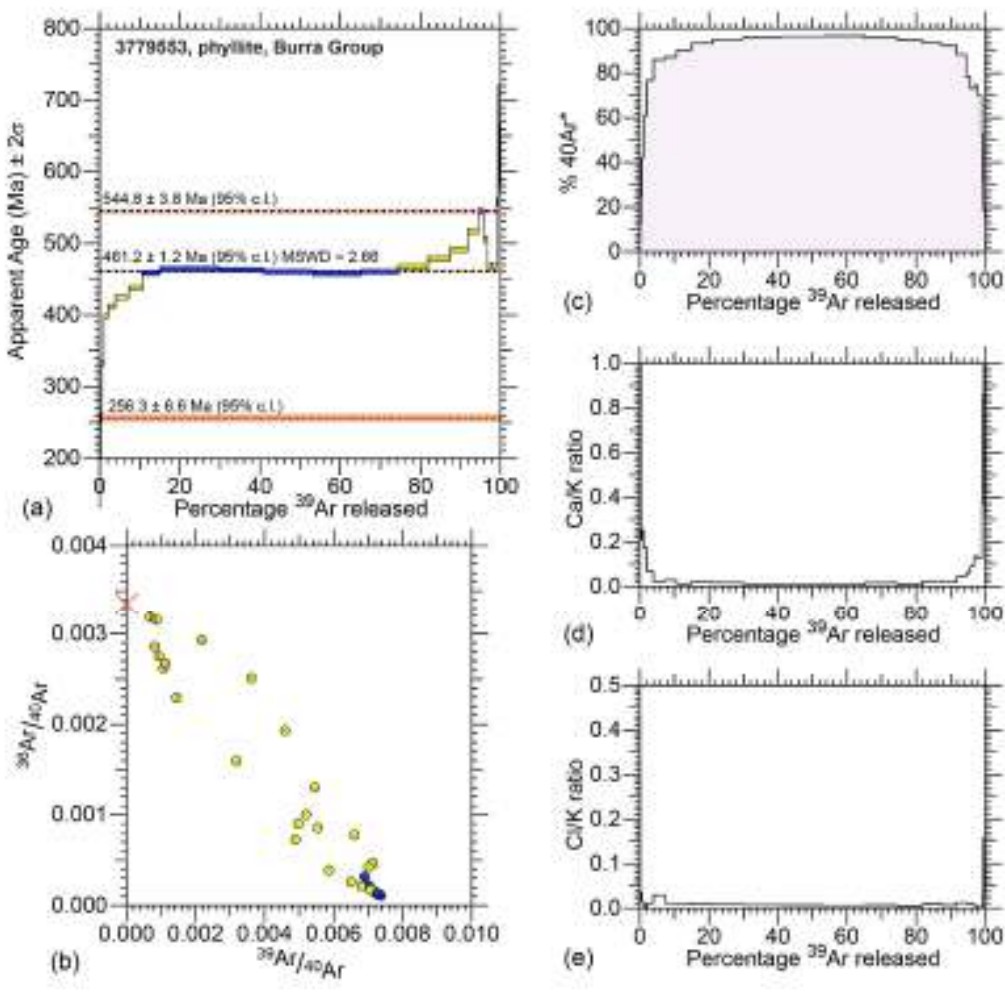

ANU CAN #36, Name: 3779553, Field ID: WPT 10, Foil: M21, Mineral: White Mica, Mass: 9.2mg, Steps: 30

Australia; SA; Burra Group; The Rocks; White-mica aggregates; clean fraction; Grain Size: 420-250micron

Plateau age 461.2 ± 1.16 Ma (95% c.l.)  Selected step(s):  10 11 12 13 14 15 16

Pearson's chi statistic 1.63 with 95% confidence range l=6 [0.18 - 1.65]

Lower limit age 256.3 ± 6.55 Ma (95% c.l.)  Selected step(s):  4

Upper limit age 544.9 ± 3.79 Ma (95% c.l.)  Selected step(s):  21

**Fig. C1. Sample 3779553 (a) age spectrum, (b) isotope correlation diagram, (c) percentage radiogenic [40]Ar ([40]Ar*), (d) Ca/K ratio, (e) Cl/K ratio.**

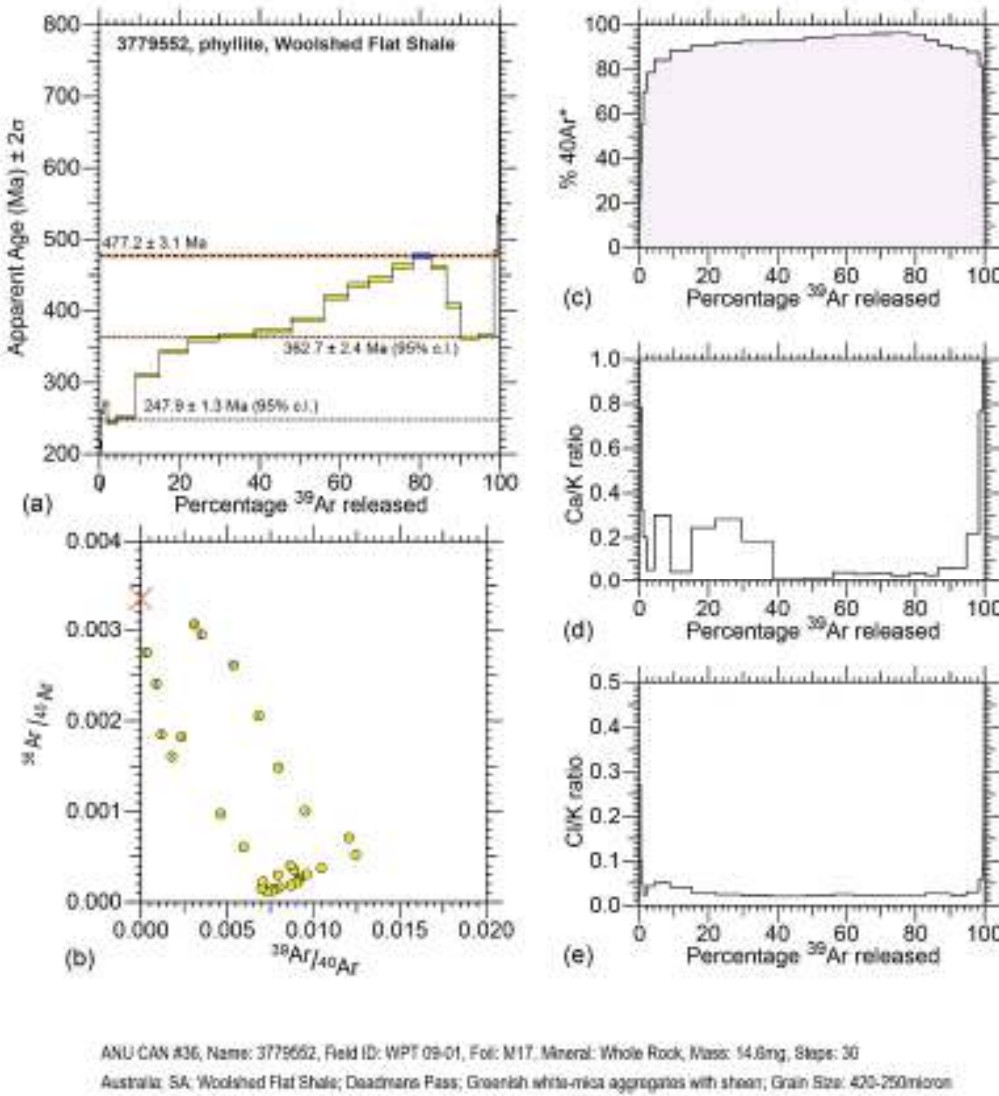

ANU CAN #36, Name: 3779552, Field ID: WPT 09-01, Foil: M17, Mineral: Whole Rock, Mass: 14.8mg, Steps: 30

Australia; SA; Woolshed Flat Shale; Deadmans Pass; Greenish white-mica aggregates with sheen; Grain Size: 420-250micron

Upper limit age 477.2 ± 3.05 Ma (95% c.l.)   Selected step(s):  19

Age based on selected steps 477.23 ± 3.05 Ma (95% c.l.)

Lower limit age 247.9 ± 1.25 Ma (95% c.l.)  Selected step(s):  7 8

Intermediate asymptote age 362.7 ± 2.41 Ma (95% c.l.)  Selected step(s):  22

**Fig. C2. Sample 3779552 (a) age spectrum, (b) isotope correlation diagram, (c) percentage radiogenic $^{40}$Ar ($^{40}$Ar*), (d) Ca/K ratio, (e) Cl/K ratio.**

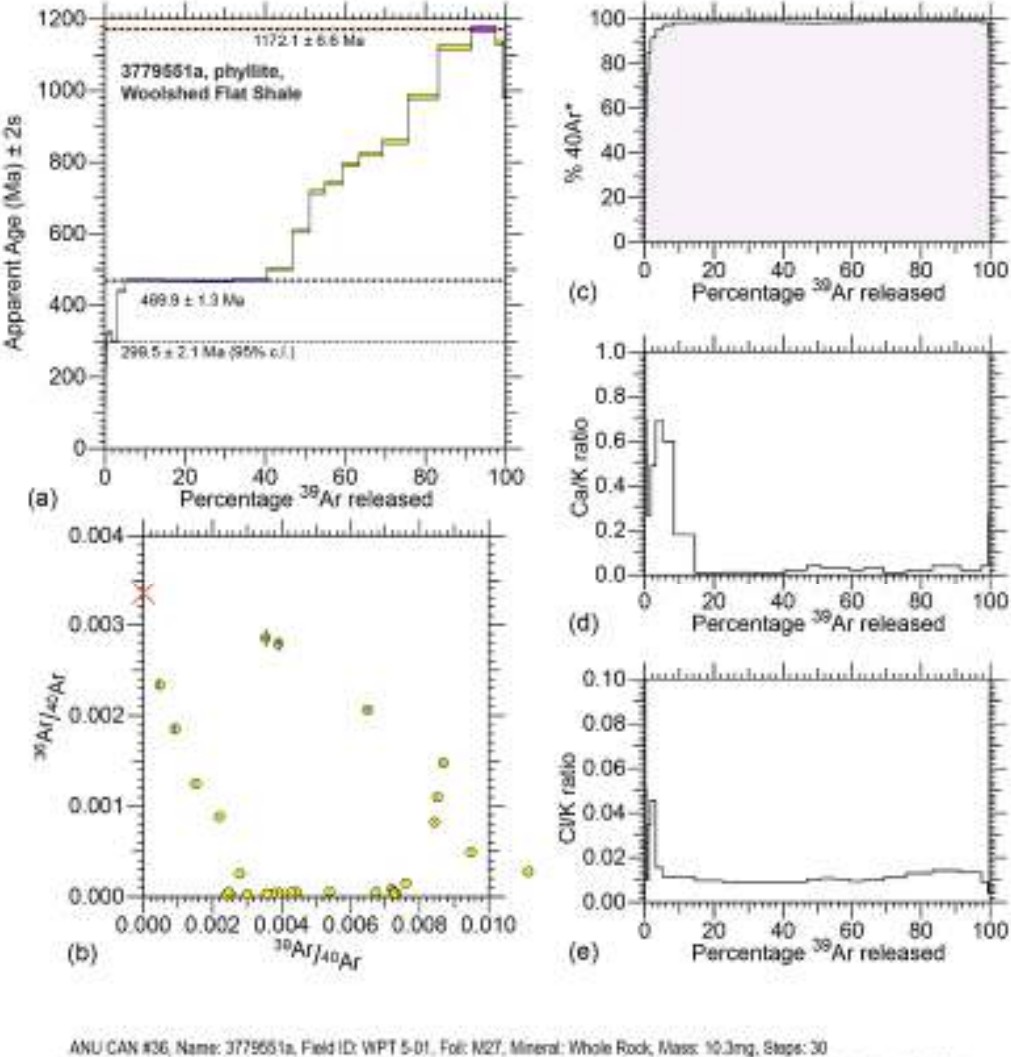

ANU CAN #36, Name: 3779551a, Field ID: WPT 5-01, Foil: N27, Mineral: Whole Rock, Mass: 10.3mg, Steps: 30
Australia, SA, Woolshed Flat Shale; Torrens Gorge; Mixture with greenish white-mica crystals and shiny quartz; Grain Size: 420-250micron

Uppper limit age 1172.1 ± 6.6 Ma (95% c.l.) Selected step(s): 24

Pseudo-plateau age 469.9 ± 1.3 Ma (95% c.l.) Selected step(s): 10 11 12 13 14
Pearson's chi statistic 0.92 with 95% confidence range f=4 [0.11 - 1.63]

Lowqer limit age 299.5 ± 2.1 Ma (95% c.l.) Selected step(s): 8

**Fig. C3. Sample 3779551a (a) age spectrum, (b) isotope correlation diagram, (c) percentage radiogenic $^{40}$Ar ($^{40}$Ar*), (d) Ca/K ratio, (e) Cl/K ratio.**

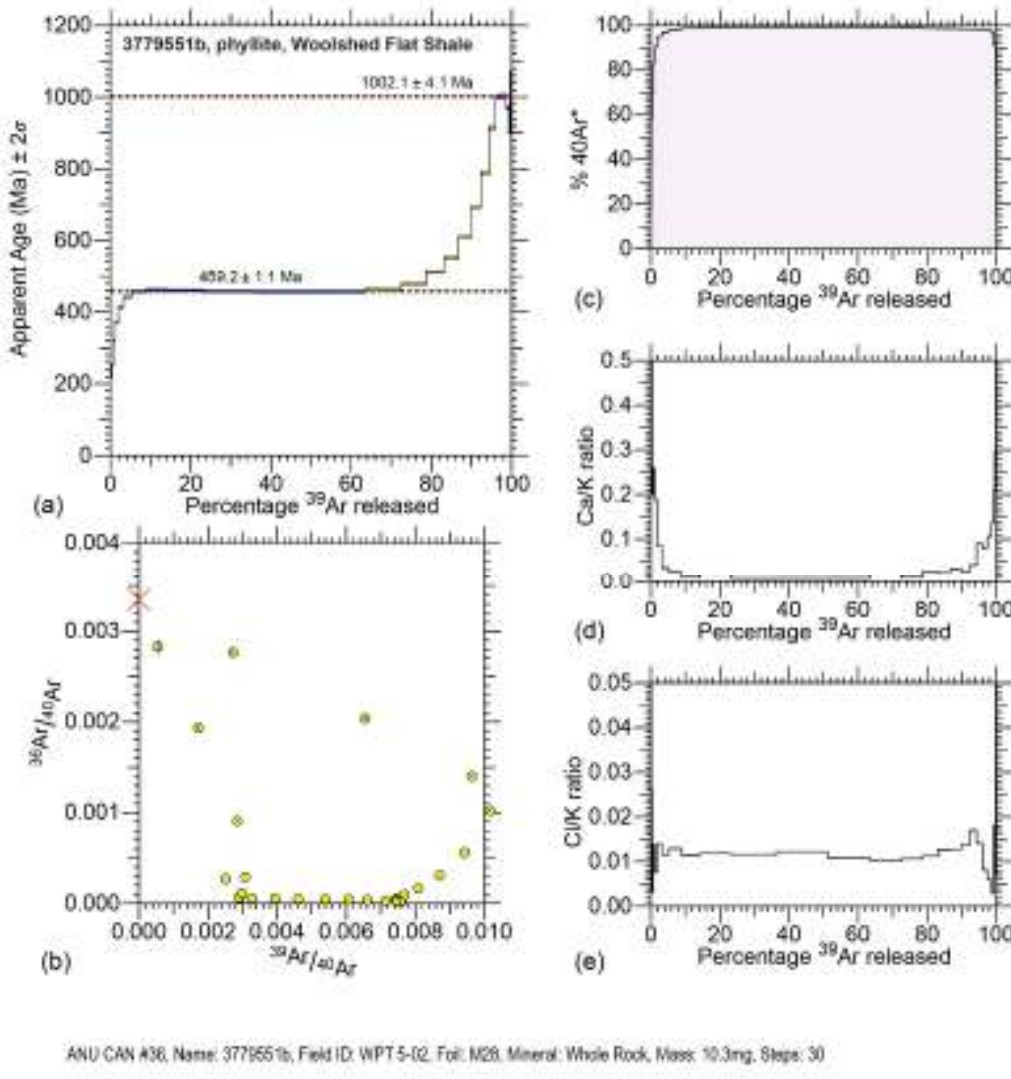

ANU CAN A36, Name: 3779551b, Field ID: WPT 5-02, Foil: M28, Mineral: Whole Rock, Mass: 10.3mg, Steps: 30

Australia; SA; Woolshed Flat Shale; Torrens Gorge; Greenish white-mica aggregates with sheen;
Grain Size: 420-250micron

Upper limit age 1002 ± 4.1 Ma [95% c.l.] Selected step(s): 23,24

Plateau age 459.22 ± 1.11 Ma [95% c.l.] Selected step(s): 9 10 11 12 13 14
Pearson's chi statistic 1.65 with 95% confidence range f=6 [0.18 - 1.65].

**Fig. C4. Sample 3779551b (a) age spectrum, (b) isotope correlation diagram, (c) percentage radiogenic $^{40}$Ar ($^{40}$Ar*), (d) Ca/K ratio, (e) Cl/K ratio.**

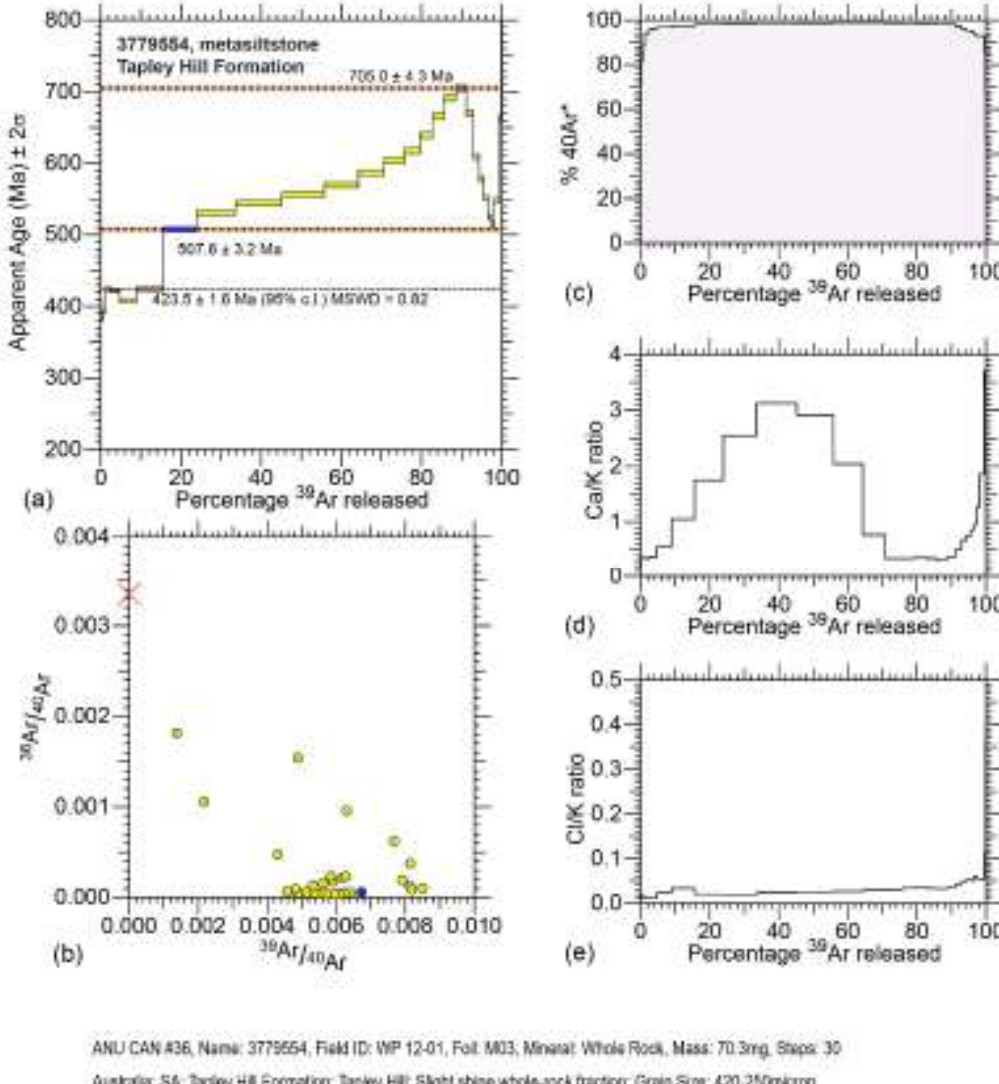

ANU CAN 436, Name: 3779554, Field ID: WP 12-01, Foil: M03, Mineral: Whole Rock, Mass: 70.3mg, Steps: 30

Australia; SA; Tapley Hill Formation; Tapley Hill; Slight shine whole-rock fraction; Grain Size: 420-250micron

Upper limit age 705.0 ± 4.33 Ma (95% c.l.) Selected step(s): 20

Intermediate limit age 507.6 ± 3.23 Ma (95% c.l.) Selected step(s): 9

Lower limit age 423.5 ± 1.59 Ma (95% c.l.) Selected step(s): 5 6 8

**Fig. C5. Sample 3779554 (a) age spectrum, (b) isotope correlation diagram, (c) percentage radiogenic $^{40}$Ar ($^{40}$Ar*), (d) Ca/K ratio, (e) Cl/K ratio.**

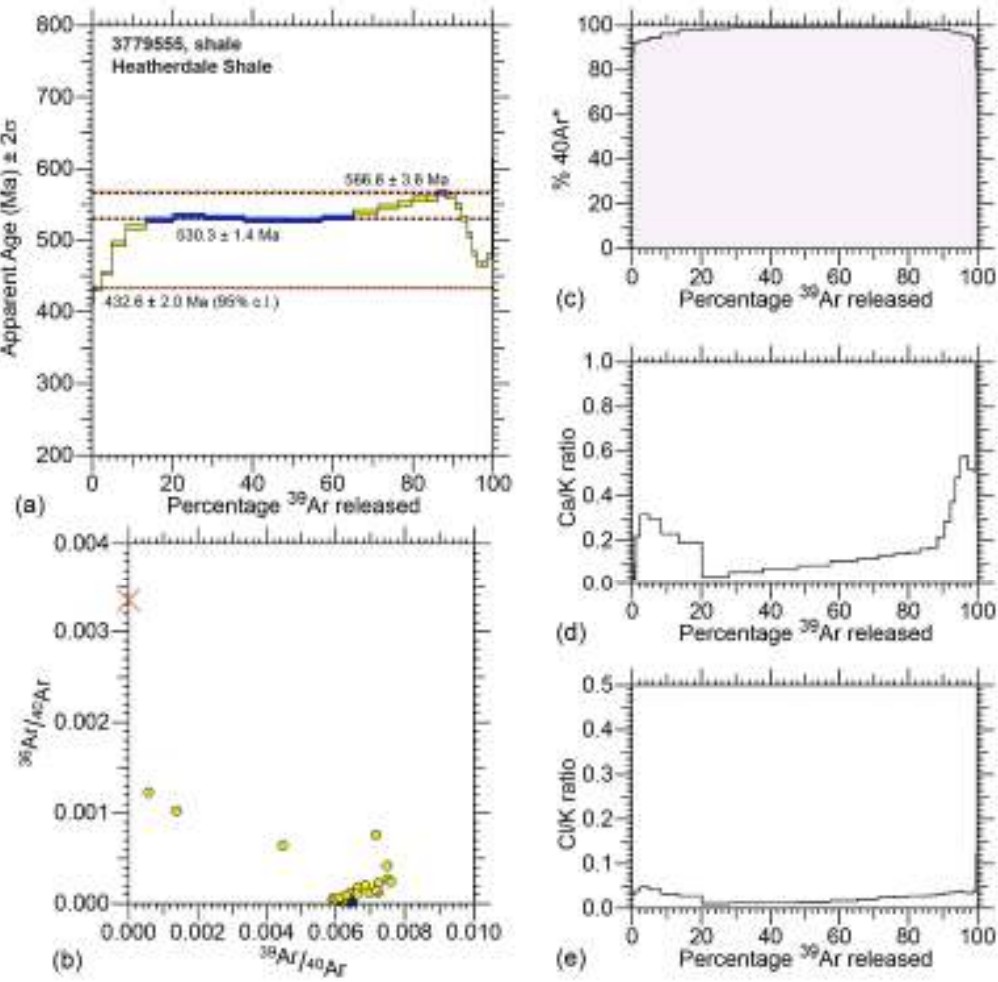

ANU CAN #36, Name: 3779555, Field ID: WP 13-01, Foil: M11, Mineral: Whole Rock, Mass: 100.4mg, Steps: 30

Australia; SA; Heatherdale Shale; Sellick Hill; Dark crystals with sheen; clean fraction; Grain Size: 420-250micron

Upper limit age 566.6 ± 3.67 Ma (95% c.l.) Selected step(s): 19

Plateau age 530.3 ± 1.37 Ma (95% c.l.) Selected step(s): 8 9 10 11 12 13
Pearson's chi statistic 1.52 with 95% confidence range t=5 [0.15 - 1.64].

Lower limit age 432.6 ± 2.00 Ma (95% c.l.) Selected step(s): 3 4

**Fig. C6. Sample 3779555 (a) age spectrum, (b) isotope correlation diagram, (c) percentage radiogenic $^{40}$Ar ($^{40}$Ar\*), (d) Ca/K ratio, (e) Cl/K ratio.**

## Author contributions

Study design and field work by AR, MF, WP, SC. Sample preparation and analyses by MF, DV, GL, NG, AC. Interpretation and data analysis by all. AR prepared the manuscript with contributions from all co-authors.

## Competing interests

The authors declare that they have no conflict of interest.

## Acknowledgements

This work has been supported by the Mineral Exploration Cooperative Research Centre whose activities are funded by the Australian Government's Cooperative Research Centre Program. This is MinEx CRC Document 2022/27. The work is also supported by the Geological Survey of South Australia. Georgina Gordon is thanked for assistance with HyLogger™ analysis at the Department for Energy and Mining's South Australian Drill Core Reference Library, Adelaide. This study has benefited greatly from insightful reviews by Jörg Pfänder and an anonymous reviewer.

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
