# Peer review of "Complex 40Ar/39Ar age spectra from low-grade metamorphic rocks: resolving the input of detrital and metamorphic components in a case study from the Delamerian Orogen"

_Geochronology, 2021_

## Author Response (AR1)

**Author Comments (gchron-2021-41)**

**Title:** Complex 40Ar/39Ar age spectra from low metamorphic grade rocks: resolving the input of detrital and metamorphic components in a case study from the Delamerian Orogen

**Authors:** Anthony Reid, Marnie Forster, Wolfgang Preiss, Alicia Caruso, Stacey Curtis, Tom Wise, Naina Goswami

**General response**

We thank the reviewers for their time in reviewing our manuscript. We respond to their comments below and are very keen to provide a revised version to the journal.

In preparing our revision, should we be invited to, we would like to add two other authors to our list of authors, being our close colleagues within the ANU Argon laboratory. These two colleagues have been instrumental in developing the software that we have used as well as helping us with compiling and developing an expanded methodological description as has been requested by the reviewers. Because of the level of detail they have been asked to go into to help us prepare both the following response as well as the paper itself, we believe that co-authorship is warranted for them.

**Referee Comments 1 – Anonymous**

Anonymous Referee #1, 21 Jan 2022

Reid et al. present new mineralogical and 40Ar/39Ar (furnace step-heating) data from fine grained, low-grade metamorphic rocks of the western Delamerian Orogeny, and use that data to distinguish between detrital and neoformed mica components. The data are also used to propose simple thermal history interpretations. The data are valuable, although the data presentation is incomplete, and I suggest that other interpretations can be reached, which have not been sufficiently explored.

My main concern is the 40Ar/39Ar data are interpreted solely considering combinations of the time of mica formation in a source region, the time of recrystallisation in the Delamerian Orogeny, and possible Ar loss via thermally driven diffusion during post-Delamerian tectonics. The logic behind the interpretation is accurate relative to these three issues. However, the authors should also consider the highly significant effects of secondary alteration, especially given the mineralogical data that is presented, which surprisingly is not considered in the interpretation section of the Ar isotope data. A robust approach would be to use mixing lines in ternary space to delineate, for example, between combinations of radiogenic 40Ar, 39Ar(K), 37Ar(Ca), 38Ar(Cl) (although unfortunately the samples were irradiated in a Cd-shielded position). These could be combined with the documented mineralogy (given the lack of microprobe data) to at least draw conclusions about which reservoirs are contributing to each heating step. This is fairly commonly done (e.g. see Challandes et al., 2008; Popov et al., 2019), and avoids ignoring fluid-related effects. E.g., were the protolith, Mesoproterozoic micas not altered at all between their original formation and deposition approximately 500 Ma later? Are the Delamerian-aged metamorphic micas unaltered? Has alteration reduced their age relative to their time of formation?

This is very helpful commentary. In addition to the York Plots, we have plotted 39Ar(K), 37Ar(Ca), 38Ar(Cl) and are interrogating what this might mean for the presence of alteration in the samples. A plot showing radiogenic 40Ar (*40Ar) is now included which shows the *40Ar for each step of the experiment. In addition, a ternary plot as suggested above has been made to be included. These analyses will be incorporated into the revised paper.

The 40Ar/39Ar methodology and data presentation are incomplete (see the recommendations in Schaen et al. (2021); https://doi.org/10.1130/B35560.1):

The methodology and data presentation that are recommended by Schaen et al 2021 are all addressed and available for our experiments. If these were not in the present supplementary data and/or within the paper, then that will be remedied. An electronic check list is available for this information and it will be included in the submission.

1. Line 174 reports that the samples were wrapped in Al foil, whereas line 481 reports they were wrapped in Sn foil.

This is a misunderstanding can be easily fixed. The samples were wrapped in Al foils for irradiation, rewrapped in Sn to be analysed in the mass spectrometer.

2. Data tables are required for the analyses of the flux monitors and the K2S04 and CaF2 salts. Reid and Forster (2021) only provide data obtained from the samples.

We can provide an excel sheet with the information on the flux monitors and the K2S04 and CaF2. This is not a problem. It is interesting to be asked for this as a number of labs do not analyse any salts but use the default values for these in their data reduction. However, yes our lab has this information available and we can include it.

3. How was the mass discrimination determined? Provide the values that were used so the dates can be calculated from the raw data by the reader. Were these dependent on ion beam intensity across the range of intensities measured ?

We have accurate records of the calculation of the Mass Discrimination value. All the Air Shot data can be provided that relate to these samples. We do up to 5 Air Shots on each side of samples which are analysed in the furnace. All this information can be included in the submission. I have not seen other publications where significant detail has been included on this.

4. Which collector configuration was used (Argus VI), and what were the Faraday/CDD yields (if appropriate)?

We have diverse ages of samples that are analysed in our laboratory, however we do not work with very young samples (e.g. rarely <1 Ma), so we use all Faraday Cups very successfully. Cross calibrations are done on the Faraday Cups regularly. We can provide this information in the submission, include a range of yields for these samples if that is required. We calculated the expected range of yield on each sample prior to their preparation, so as to maintain quality results from the mass spectrometer.

5. I gather from the text that the irradiation package included the flux monitors in a linear stack. What was the distance between each flux monitor?

Yes, we have a glass tube and the foils are placed or stacked meticulously into the tube. Of course, we have details on the configuration of every irradiation, in this particular irradiation the samples were between flux monitors spaced at 6-7mm apart. For each of the flux monitor foils, between 5-8 separate analysis were done.

6.      Provide a description of how the blanks          for individual heating steps were determined once the sample was already loaded into the furnace crucible .

All backgrounds are subtracted for every step. As the reviewer will be aware, blanks cannot be done once a sample is dropped as blanks are done on increasing temperatures which would not be possible. Blanks from the furnace are monitored carefully. Blanks are done to monitor the isotope levels prior to each sample for a range of temperatures from 350°C to 1450°C. A full set of blanks are repeated at the end to monitor changes. Blanks are also done in-between samples in a set if it is regarded necessary. All data are recorded.

Specific points

Introduction: The Introduction is clearly written although it would benefit from some more specific aims that are relevant for the Delamerian Orogeny. E.g., the info in line 108 would be useful in the Intro.

This can be incorporated.

Line 108:  Where does this 480 Ma come from? The only previous reference to 480 Ma is in line 76 (Ar/Ar laser analysis of hornblende and biotite; Turner et al., 1996). Are the authors suggesting that the youngest of these Ar dates was previously taken to signal the end of orogenesis?  The Ar dates are from granites so surely they just record cooling, and not crystallisation. U-Pb dates would be more reliable. Are there any? Line 69 suggests orogenesis was over at about 490 Ma.

This requires some clarification in the revision. The U-Pb dates are as young as c. 485 Ma, while the available $^{40}Ar/^{39}Ar$ dates are mostly around c. 480 Ma.

The Hylogger results presented in Appendix A require more clarity. As described in Appendix A, each sample is divided into a series of runs. For each sample, there are separate bands, each of which  consists of three rows of coloured squares, which correspond to mineral compositions obtained by either SWIR or TIR. How should these three parallel rows be interpreted? Is each parallel row, one traverse such that each band of three parallel rows is three traverses? Or, is one band of three parallel rows, one traverse?

The HyLogger collects both SWIR and TIR data simultaneously.  Therefore, the three rows of data presented for each sample in Appendix A relates to a single traverse and each sample was traversed five times.  The three rows of data represent SWIR Mineral 1, SWIR Mineral 2 and TIR Mineral 1.  However, we can see how this will be confusing for a reader unfamiliar with HyLogger data, so these Appendix figures will be edited to remove this part of the figure and simplify the results for the reader.

Furthermore, the colour for Mg-Chlorite is extremely similar (almost not distinguishable) from the colour for albite.

These are the standard colours used by the Geological Survey of South Australia and are best viewed on screen in the Hylogger interpretation software. The colour for Albite and Chlorite-Mg are similar and the figures in Appendix A will be edited to ensure these colours do not appear adjacent to each other in the final presentation.

Lines 136 – 137. The TIR also shows a considerable amount of muscovite.

Yes, the TIR does also indicate muscovite in this sample, however depending on the mineral group, the spectral information is more definitive in one wavelength region compared to another depending on the particular absorption feature.  Previous research has detailed the various bonds and overtones measured by the HyLogger.  In this case, the Al-OH bond, represented by the 2,200 nm absorption feature, is well

established in the literature to be the main diagnostic feature for white mica (muscovite/ phengite etc.). The absorption features for white mica in the TIR occur at approximately the same wavelength ranges as other mineral group absorption features such as various silicates (approx. ?? nm).

Typos in lines 129, 273, 319, 407

These will be fixed.

The plots in Fig 5 can be improved. Use the full space of the graphs. E.g. in 5a, modify the date axis to range between 200 and 650 Ma (the space between approx.. 750 Ma and 1200 Ma is currently blank).

All of the age spectra are on the same Y-axis scale to allow easy comparison between them. The reviewer's comment would mean that the different spectra could not be visually compared as easily and our preference is to leave the Y-axis at the same scale.

To help the reader, I suggest colour coding segments of the age spectra and York plots so the reader can match the topology of the spectra with specific trends in the York plots that are referred to in section 5. E.g. line 197. Which final steps, and which points are these on the York plot? Line 213 – which are the beginning and the final steps in the York plot? Etc………

We have updated software that colour codes the steps used. In addition this updated version also includes the errors on the York plot and step numbers can be included. This will be in the revision.

What is the author's definition of a minimum date? In some cases, the minimum date seems to be the 40Ar/39Ar date of the first heating step (e.g. 3779555), whereas in other cases the date of the lowest-T heating step is ignored (e.g. sample 3779551a). If significance is attached to a minimum date, then describe how the minimum date is defined.

We will go over the minimum date for each of the spectrum. We will define what this term means and keep a continuity in the use of the term.

Lines 240-242. Theoretically, a range in diffusion characteristics in a single crystal would yield a staircase spectra from a single crystal , depending on the t-T. However, complex age spectra can also be obtained from "single crystals" because they are frequently polycrystalline with perhaps several generations that arise by deuteric and low-T alteration (e.g. K-feldspar from the Chain of Ponds, Chafe et al., 2014; Klokken K-feldspar, Parsons et al. 2013; Itrongay K-feldspar, Popov et al., 2020; muscovite from Larderello, Italy, Bulle et al., 2020).

Yes, we agree with this. Comment on this will be incorporated.

Line 254. What do the authors mean by "main mineral gas reservoir"? Any combination of reservoirs that don't contain an initial component? There may be several radiogenic reservoirs if the grain is not 100% monomineralic.

This entire paragraph can be removed from the Discussion. Nevertheless, it is important to recognise however, that age spectra are the result of mixing between different gas reservoirs and in that sense all of the metrics collected in the step heating experiment are not "pure samples" of individual gas components. We will re-phrase this to make it clearer. It would read better as:

"An example of white mica from the South Cyclades Shear Zone, Ios, shows the initial heating steps having a distinct isotopic composition from the majority of the heating steps, representing a mixing between atmospheric and radiogenic gas compositions within the mineral separate (Forster and Lister, 2009)."

Lines 256 – 257. I realise this refers to previous work, but what is the justification for the statement that the older, high-T steps are infected with excess 40Ar? Could this also be inherited Ar from a xenocrystic component? More robust justification is required.

The York Plot for these samples shows the degree of mixing between atmospheric, radiogenic and excess argon, by definition. The steps we refer to plot towards the excess Ar composition on the York Plot. As indicated above, however, this entire paragraph can probably be removed from the Discussion.

Is 39Ar recoil evident in some age spectra? Several age spectra have a sudden reduction in date in the higher-T steps, followed by a subsequent increase in phyllite 3779553. Is this a recoil effect, which would not be surprising given the extremely fine grained nature of the groundmass? In this case, could the older dates (e.g. 709 Ma for sample 3779554) be artificially too old? Or, is a younger, more retentive phase being degassed? This should be addressed.

This is an interesting thought that could be considered due to the fine-grained nature of the samples. Where we do find recoil, it has been detected in the Arrhenius plot. In addition, a sudden change in %*40Ar can also have such an affect, and as we now include a plot of the *40Ar release across the experiment this can easily be seen if this is causing an effect as described here. This will be looked at in our revision.

I am not convinced that the 40Ar/39Ar date of 511±2 Ma (3779554) accurately records the timing of cleavage formation . Before making such a statement, the authors should demonstrate that this single heating step did not release Ar from any secondary alteration phases. This sample also hosts microcline, calcite and albite. E.g. did this step liberate any 37Ar from the calcite, and what was its influence on the 40Ar/39Ar date? Does this step contains and K released from the microcline? What is the relationship between the microcline and the foliation?

This is a fair comment. A more accurate statement would be that the 511 Ma age must be an upper limit on the timing of cleavage formation. The cleavage could have formed at that time, or sometime after. The c. 511 Ma age represents a limit on the age spectrum in the sense that the spectrum is a mixing between two or more age components in the rock.

The age spectrum is corrected for any interferences from interfering isotopes e.g. 37Ar, thus avoiding any effects of $^{37}$Ar on the $^{40}$Ar/$^{39}$Ar date.

Lines 314 – 324. Statements about the tectonic setting during deposition can only be made if it can be confidently established that the oldest 40Ar/39Ar dates are accurate measurements of the age of the muscovite grains. This has not been demonstrated for all of the analysed samples. This process will always be best with U-Pb concordia ages, but it is less clear when 40Ar/39Ar (single step dates!) are used. I have no issue with the logic in this paragraph, although I suggest the authors at least acknowledge this, or make a more detailed comparison between any detrital zircon U-Pb data, and their oldest 40Ar/39Ar step dates. For example, did the Mesoproterozoic grains undergo any secondary alteration before they were deposited and captured  in the Delamerian Orogeny? These statements should be addressed prior to making tectonic interpretations of single step-dates.

A modification to acknowledge the uncertainties in the interpretation will be incorporated.

Lines 337 – 359. The authors account for their Ar data by combining i) the time of growth/deformation, and ii) Ar loss by diffusion, which is subsequently related to exhumation. Given the mineralogy of the rocks (and the Hylogger analyses), the authors should also address the possibility of a reduction in date relative to the timing or deformation caused by fluid flow events that post-date Delamerian deformation. Could alteration be responsible for the reduction in date? If not, then why not, and use the available data to show this. The authors attach significance to the younger step-dates in sample 3779552. This sample hosts chlorite and

albite, with a very low amount of muscovite. Could these younger dates simply be a result of secondary alteration? This should at least be addressed and a more robust justification is required to interpret the dates in terms of t-T paths, exhumation, fault reactivation .

Of all the samples, 3779552 is probably the best example of how alteration could be playing a role in the age profile. We acknowledge the submission did not adequately cover the likely effect of alteration and this can easily be incorporated into the revision.

**Referee Comments 2 – Jörg Pfänder**

Metamorphic rocks are an integral product of a series of complex atomic to orogen scale processes that operate over geological times sequentially and parallel at variable but mostly unknown physico-chemical and thermodynamic conditions. Deciphering the timing of these processes (i.e. discrete and geologically significant age signals) from them is a highly challenging task in geochronology. The manuscript by its title intends to contribute to this issue and takes efforts to demonstrate that step heating of low-grade metamorphic rocks (and in part mineral fractions) can provide valuable information on the age of detrital components as well as on the timing of the low-grade metamorphic processes.

Unfortunately, this goal is not reached and the manuscript does not provide convincing evidence that the detrital and metamorphic age signals can be reliably resolved.

We find this statement surprising. The fact that the rock depositional age is known from previous work and that some of the $^{40}Ar/^{39}Ar$ ages in our age spectra are older than this means that they must be derived from detrital minerals in these low metamorphic grade rocks. Alternatively, the older ages are a result of excess 40Ar. However, the York Plots and data tables themselves demonstrate that the main age patterns are from heating steps with little or minimal excess 40Ar. The simplest interpretation is that these age spectra represent mixing between a detrital component and a metamorphic component.

Although the data itself seems to be sound, there is a fundamental problem with the chosen methodical approach, which strictly speaking is backward -looking in that it analyzes (mostly) whole-rocks instead of carefully selected mineral separates by step-heating alone.

The point of whole rock analysis is that constraints on the tectonics/alteration/regional geology and so on, can be obtained even in situations like ours in which the rocks are too fine grained to be able to provide mineral separates. An alternative would be to undertake ultra-fine grained mineral separations such as by centrifuge, however this is not our preferred method. The study deliberately attempted to date phyllitic rocks and other low metamorphic grade rocks to determine what results could be reliably determined from them. We believe this is a useful contribution.

This approach has been chosen to circumvent the problem of recoil Ar loss during irradiation, which in principle is reasonable, but even pure mineral fractions having a well defined grain size spectrum often provide 'complex' age spectra during multi-grain step heating (even for single grain step heating), which results from frequently present excess argon and argon loss, controlled by thermal overprinting, recrystallisation, fluid-inclusions and/or intra-grain deformation. Therefore, without additional constraints, such as ages from other geo-/thermochronometers obtained on the same samples, or other Ar-Ar approaches such as single grain total fusion, single and multi grain step heating or in-situ dating, the gained age information remains unconstrained. This drawback has been already stated by Forster & Lister (2004) who introduced the chosen methodical approach: 'Frequently measured ages (FMAs) in individual datasets can then be recognized using statistical analysis. The significance of FMAs must be independently assessed'.

This statement emphasizes the importance of doing statistics on a significantly large dataset, and the importance of independent evidence. None of both has been sufficiently well addressed in the study.

In an ideal world we would have collected and analysed many more samples. However, the sample set covers a region of interest in the Delamerain Orogen, and therefore provides a significant improvement to the geochronology in this region (ie prior to this study there was effectively no $^{40}Ar/^{39}Ar$ analyses in this region).  The alternative methods suggested by the reviewer would mostly not provide more information, in fact would lose information, for example single grain total fusion or in-situ dating where information would be mixed and would not provide the information the review is suggesting.

The manuscript is also not in a shape as is expected for an international journal regarding the overall structure and writing. This, for example, refers to the organization of the samples and results chapter (sample by sample descriptions), but also to data documentation and presentation, which are insufficient .

We are uncertain why sample-by-sample treatment of data means the paper is unsuitable. How would the reviewer suggest we approach describing the results? It seems a fairly reasonable thing to do. However, we will review this.

The data further miss a thorough and careful in-depth evaluation (inverse isochron ages and intercepts of sub-datasets used to derive ages, evaluation of contributions from  Ca-, K- and Cl-derived Ar-isotopes), which in particular is required given the uncommon approach of doing step-heating experiments on whole rock low-grade metamorphic rocks  that contain high amounts of chlorite. Such complex samples require a more critical and sophisticated data evaluation and discussion.

The discussion is weak, not very well written, and far away from being acceptable for an international high-quality journal. It mostly remains vague, is often hard to follow, not to the point, full of unsupported and confusing statements, and contains passages that are rather empty phrases than robust scientific arguments  (example: 'Complex age spectra and multiple gas domains are present in our samples from the low metamorphic grade rocks in the Delamerian Orogen. The presence of upper age asymptotes that also have distinct isotopic compositions, is consistent with those ages being from distinct gas reservoirs. Taken together, the complex age spectra in this study clearly represent the mixing between older detrital components and younger age components ).

All the information on isotope interferences are done and are reported on in our Supplementary Data. We have included an expanded section on data quality in the revision.

The study hampers another fundamental but rather common problem, namely that the geologists among us often desire to have age information from rocks that these rocks and the available methods can not provide in an easy, fast, cheap and simple way due to fundamental physical limitations. Working on such (mineralogically) complex rocks with such a complex history (by far too many degrees of freedom) requires tremendous methical efforts and an in-depth evaluation of the gained data by considering physico-chemical principles. Just plotting 'easy to read' age spectra plots and deriving obvious age information is not sufficient .

This is certainly the case. And yet, geology progresses one analysis at a time. If we waited until we had a 'perfect' analytical data set, our results would never be published.

This manuscript is an attempt to publish step heating experiments on a series of deformed rocks. It is not intended to be the most exhaustive study of the mineralogy, chemistry, fluid inclusions and so on that could be done.

More importantly however, our main point is that complex age spectra have meaning and can be interpreted with ease using the concept of asymptotes and limits discussed by Forster and Lister (2009 and other publications).

Necessarily the results here have been given interpretations. We will work through the revision to reign in some of our 'enthusiasm' for the simplicity of some of the interpretations. More qualifying statements are required acknowledging the reconnaissance nature of the data. In addition, some of the samples could well be suffering due to later alteration and this is something we will look at more closely in the revision.

In summary, the manuscript needs substantial and very comprehensive revisions. A more comprehensive and systematic evaluation and presentation of all data as well as a clear and concise discussion of the results is required, along with a much better consideration of existing (methodical) literature. Doing so, and comparing the results with existing literature data, there might be a valuable scientific outcome with some importance for the community working on the evolution of such rocks and the controlling geological processes.

We will incorporate some of the methodological literature, noting the excellent, specific examples given to us by Reviewer 1.

Chapter related comments

The abstract is in parts too vague. For example, low grade rocks also contain detrital minerals that have been overgrown during metamorphism. Then it becomes not clear that whole-rocks have been analyzed, as likely most readers will assume that a common approach has been applied where mineral separation precedes dating.

This can be rectified with a comment on the nature of the samples analysed.

Complex age spectra are interpreted as consisting of several age components. This is misleading, as it implies that discrete and real age events are present. An 'age component', however, might simply result from mixing of gas fractions from different (disturbed) grains with different ages, size and/or Ar release properties, and thus is not an age but an artifact. Would suggest to term them 'virtual age components'. This then might question the subsequent interpretation, where neither the problem of mixing of different gas fractions, nor the potential presence of excess argon or argon loss during reheating or recrystallization have been obviously addressed. This needs to be done.

Line 12. 'Low metamorphic grade rocks' need to be 'Low grade metamorphic rocks' .

Ok.

The introduction is too short and misses a brief review of methods and case studies that have addressed dating of low-grade metamorphic rocks. Such a brief review should follow the first paragraph, which ends with a statement that it is difficult to separate different age signals. This is true, but several approaches likely exist and these should be briefly mentioned (K-Ar dating of shales, clay- and siltstones, etc.; Ar-Ar dating approaches such as in-situ, single-grain total fusion, etc.).

We can expand upon some of the references we cited as previous examples of work on low metamorphic grade rocks.

Line 29-30. Sentence does not make sense.

Line 29-30. This sentence seems somehow to end strangely.

Will review.

Line 40. … provide information on the age of                  possible ….

Thank you. Helpful, specific advice. We will change the sentence to this.

Line 43. What do you mean by in-situ laser microprobe methods? Laser ablation in-situ Ar-Ar dating? Then this should be outlined more clear ly.

Yes, this is what we mean. We will add clarification.

Line 46. 'Age populations' seems being misleading. Such commonly result from a large number of samples with a single age, or from single-grain total fusion experiments on a single or of multiple samples, and I doubt that sufficient ages can be derived from five whole-rock samples by a step heating approach to resolve statistically significant age populations.

And yet, these are the data we have available. In our revision we will provide a cumulative probability plot of all of the ages obtained in the study to clarify the more significant age populations across the samples analysed.

The geological setting chapter should be shortened by at least 15-30% , it seems a bit too comprehensive given that the goal of the paper is mostly a methodical one. Reduce the text to what is needed to understand where the samples are from and how they evolved. Rock types should be mentioned from time to time instead of only unit names (line 40-41).

The geological setting can be reduced.

Line 66. Is there a word missing after metamorphism?

Yes, "at".

Line 73. Do you mean a thermal laser? I.e. laser induced step heating ?

Yes.

The sample descriptions chapter is not straightforward to read, as it is organized in form of a sample by sample description (and in part contains useless information: e.g. line 130: 14 km east-north-east of Adeleide). The chapter should be condensed by summarizing the major features present in all samples (including the constituting minerals as these are mostly the same, just note if some untypical features/minerals are present) and by outsourcing information in a table (as in part has been already done in Table 1, why describing places in the text when coordinates are given?). The images in Fig. 4 are not very helpful, thin-section images would be much better. Either replace the field images by thin-section images, or complement them by the latter.

We can reduce the descriptive parts of the section. However, because we have a relatively small data set, we prefer to discuss the results sample by sample. This is because there are important differences between each sample that merit explicit consideration. Thin section images are worse than the field images. The samples are very fine grained.

Line 148-149. Very near: How near?

Within 15m. We will clarify this.

As the sample description chapter, the Ar-Ar results chapter describes the data and/or plots in a quite monotonous manner sample by sample. As said already, this is rather annoying to read. The authors should reorganize it, and instead should work out the general features and describe them once, and then, supported by examples, should describe the important and significant aspects of the data.

In our view, geochronology papers should have a           section that simply describes the results, in this case sample by sample. Any section that then describes general features is something that should be tackled in the Discussion, which we have attempted to do.

All ages that were derived from the age spectrum plots and which are assumed to be geologically meaningful need to be reported in a table  along with all associated data (errors, 40Ar/36Ar intercepts, MSWD, etc.; see Schaen et al., 2020; Renne et al., 2009).

We will expand Table 2 to add some of the statistical metrics for the interpreted ages. Schaen et al 2020 and Renne et al 2009 have been used as a method to report our supplementary data. All recommendations have been adhered to where applicable.

The term 'York plot' is fully uncommon. Either use inverse isochron plot or isotope ratio plot, as common in the Ar-Ar community . Ironically, the original York publications in 1968/1969 deal with linear best fit isochrons and correlated errors, a procedure that is unfortunately not applied in this manuscript, which needs to be done in a revised version.

We can re-label the York Plots as isotope ratio plot, as per the suggestion. We are aware of the issue the reviewer refers to and could go into great detail on this, however, in terms of isochrons, these are not required in this study because we recognise that we are dealing with mixtures of minerals and there is no expectation that any of the results will lie on a single isochron.

Use the term 'virtual age' instead of 'age',  as it remains finally unclear whether the derived 'ages' are of geological significance or represent averages of different events with different ages.

Does the reviewer mean 'apparent age'? We have not come across the idea of a 'virtual age'.

Line 196-199. What do you mean by a more or a less retentive domain? How does the physicochemical process of retention (i.e. diffusion) affect an isotope ratio? Not significantly to my opinion if we ignore diffusive isotope fractionation. You may tap different domains with different isotope abundances within a mineral, or different minerals at different temperatures, but just using the term 'retention' is insufficient to cover the complexity of degassing a multi-phase system.

We will remove 'retentive' from the sentence.

Line 201. The term asymptote needs  to be defined. What is the 'concept of asymptotes' as mentioned already in the abstract (line 17-18)? What you mean is simply an upper or lower age limit. So the term asymptote is misleading here as in a strict mathematical sense an asymptote is a limit to which a variable converges.

We will remove the term asymptote and instead refer to limits.

Line 209. What do you mean by the statement '36Ar/40Ar values have no atmospheric component'? This sentence is fully meaningless, if not wrong. Any gas fraction analyzed during a step heating experiment that provides detectable 36Ar potentially can contain a fraction of atmospheric argon (i.e. an atmospheric component). In principle, any datapoint in an inverse isochron diagram that does not plot on the x- or y-axis can be interpreted as representing a three-component gas mixture between atmospheric argon (plots on the y-axis at 0.0033), radiogenic argon (plots on the x-axis at its 'true' age 39Ar/40Ar) and a third component ('initial' Ar) that plots on any point on the y-axis except at 0.0033. Data scatter then simply reflects changing mixing proportions between these three 'components' during step heating.

The heating steps we are referring do indeed plot on or very near the x-axis. However, the statement should read "virtually no atmospheric component" or perhaps "very minor atmospheric component". Regardless, the statement can be omitted in the revision and not change the interpretations.

Line 212. What do you mean  by 'two domains'? Does not make sense to me.

Domains in the sense of gas reservoirs within the                composite mineral aggregates. An older component of gas, say from detrital minerals within the whole rock is one 'domain' of gas age, compared to a newly grown 'domain' of gas with a younger age. This can easily be clarified in the revision, with the word 'domain' being changed to something less specific.

The Discussion is not appropriate to an international journal. There is no but some vague geological outcome from this discussion. Although the overall approach of this study (chloritized whole-rocks) in principle is questionable, it might be worth doing such an investigation in an effort to circumvent the problem of recoil loss of 39Ar in fine grained mineral separates during irradiation. The way it has been done here, however, is insufficient as the data have not been evaluated in depth.

We have done our best to write this for an international journal. We believe that the geological meaning we can extract from the results are in fact all the more remarkable *because* we have sampled chloritized rocks. We will do our best to make the revision better.

Line 239-242. Why just different mineralogy? Mixing of the same mineral type with different ages will also provide complex age spectra. Please clarify.

Agree.

Line 243. The term crystallization in this context is misleading. A low grade metamorphic rock can't be described by a discrete 'crystallization age'. Be more clear what the maximum could be (an old detrital component).

Yes, this can be clarified.

Line 244-245. What is a geological resetting event in this context? Be more clear. And note that a 'less retentive domain' is a physical representation of a thermal 'resetting' event. So using 'or' in this context is misleading.

We will reword this statement. A resetting event could be any type of process that resets the argon age systematics within a sample. The least retentive domain is just that. It is the domain that has closed to argon diffusion last in the cooling history of a mineral or is the most easily reset by any process that causes resetting of the argon clock within that domain.

Line 254-255. What is the 'main mineral gas reservoir'? What you mean is the other endmember, the purely radiogenic component (see above). Be more clear. Why should this mixing be the result of 'deformation-induced recrystallization'? This is a fully wrong statement, as this mixing is simply due to different argon 'components' having different diffusivities (or retentivities), or different bonding energies, or whatever we want to term it. The presence of atmospheric argon and radiogenic argon has not necessarily to do with 'deformation'. The latter may trigger partial argon loss (and also uptake of argon), but this is another issue. In an inverse isochron plot, any Ar loss and any excess component force the data deviating from plotting along a straight line.

Yes this can be clarified.

Line 260. The statement 'The presence of upper age asymptotes that also have distinct isotopic compositions' is circular. The whole sentence is not understandable.

Will revise the sentence. The intention is to highlight that the older age components are distinct from younger ages in terms of their age as well as their composition (K/Ca etc).

Line 285-287. Hard to understand what is meant here.

Will attempt to revise.

Line 289. State from where the deposition age is known .

The age of deposition is given in the Geological                   Setting (and in Table 1), which the reviewer wishes us to reduce in length. We will try to incorporate some of these details into the Discussion where required.

Line 292. What are similar ages? How similar? Provide data please.

This can be clarified.

Line 303. What is the depositional age and from where is it known?

The age of deposition is given in the Geological Setting (and in Table 1), which the reviewer wishes us to reduce in length. We will try to incorporate some of these details into the Discussion to clarify.

Line 317-324. These setting inferences from very questionable 'ages' derived solely from age spectra plots obtained on very complex samples are very daring.

If required, this section can be removed from the Discussion. It is not central to the main point of the paper.

Line 353-354. Or simply a stronger resetting of smaller grains. I do not like the term 'retentivity domains'. It makes things mystic in some way, and obscures that we were dealing with a complex mixture of crystalline material. What do we physically have? We have rock fragments. Each fragment consists of a number of different mineral types, where each type has its own grain size distribution (affecting diffusivity, or retentivity). Each mineral, in the simplest case, may have its own age, but may be compromised by either Ar loss or the uptake of excess argon (or both), possibly at different times (multiple events). Clearly, this complex material and history will provide complex age spectra, but interpreting them solely by considering 'mystic' domains does definitely not do justice to the complexity of the problem. Therefore, data interpretation needs to be done in the light of what is physically present.

Yes, the resetting can be affected by grain size. We will modify the sentence accordingly.

Line 349-350. This is completely at odds. If 'ages' out of a simple sample are younger than other ages from other samples, how can this be indicative for 'a complex history of mica growth and cooling'? Such a statement is fully unsupported.

Can reword this.

Line 358-359. Useless statement.

Can delete this.

Line 372. … low metamorphic grade …

Yes.

Tables

Table 1 . First line: 515 Ma: Provide the exact datum including error (line 149) as in the second line.

Yes.

Table 2 provides too little information. The power of Ar-Ar is the fact that a dataset provides internal control on consistency. Just reporting ages and errors is insufficient.

More detail on the type of age and statistics will be added to Table 2.

Figures and figure captions

Fig. 1 is a good overview. Needs to be 'Damara Zambesi Orogen' not 'Darmara Zambesi …'

Yes.

Fig. 2. (a) and (b) are missing. Make it more clear that the red square in the lower left sketch outlines the region shown in (a) and (b). Enlarge the sample location dots.

Yes.

Fig. 3 should be deleted. It provides nothing necessary to support the goal of this study, and contains information that in part can be found in the text (and is mostly speculative). Three large images with geological maps and profiles in a paper with a methodical goal are too much.

The figure is intended to help the reader visualise the orogen and relate the low metamorphic grade sections to the eastern higher metamorphic grade regions. However, the figure can be deleted if required.

Fig. 5. The way the data is presented is mostly insufficient. First, the y-axis scale in some of the age spectra plots is too large, making it difficult to resolve individual steps and differences between them. Adopt this even at the expense of having the same scale in any of these plots.

We prefer to leave each in a standard view as this enables easy comparison between the age spectra. However, as Reviewer 1 and 2 have both asked for this, we will replot the age spectra.

In addition, provide in the figure caption or in the figures what initial 40Ar/36Ar has been used to correct the isotope composition of each step for initial argon in the procedure of calculating individual ages for the age spectrum plots (I guess that an atmospheric value has been used , but which one?).

Will be added to the caption.

Second, if an age is resolved from an age spectrum, an inverse isochron through the respective steps needs to be plotted along with the corresponding inverse isochron age and inverse y-axis intercept value (i.e. the 40Ar/36Ar of the initial argon). If the latter deviates from the atmospheric value, this needs to be considered in age spectrum age-value calculation and the potential causes need to be considered (except in cases where the correction can be neglected due to near-100% 40Ar*-values).

This approach is an interesting one and would be valuable were these analyses of single minerals. In this study, the maximum and minimum ages defined are often defined by single steps, or a small number of steps. As a result, it is impossible to calculate an isochron age. If the results were averages of multiple steps, i working out an isochron age from those data points would be a good idea. We will however, carefully go through our 'ages' and determine if this approach is appropriate.

Denote which steps in an age spectrum have been used to calculate which virtual age. And mark the temperature steps with numbers in both the age spectra as well as the inverse isochron plots so that one can see which step belongs to which in each plot . Mark the position of each of the derived 'ages' from the age spectrum at its corresponding x-axis position in the inverse isochron plot.

The York Plots can be replotted with the step numbers.

And include the datapoint errors in the inverse isochron plots.

The York Plots can be replotted with the data point errors, however, in most cases the error is smaller than the symbol.

Most of the upper 'age values' in the age spectra plots are hardly of any geological significance. For example, in Fig. 5c the age of 371.1 +/- 7.8 Ma. From which steps has it been derived? There is hardly anything that looks like a plateau . Do you really believe this age is of any geological meaning? Hard to believe, the steps are more or less continuously increasing. And as far as I can see, hardly any (sub-)dataset in the inverse isochron plots will provide an atmospheric intercept. Something as for example in Fig. 5d

needs to be interpreted, to my opinion the plot             suggests Ar-loss for the very low temperature steps, and lot's of excess Ar in the steps that deviate to the lower left side (same for Fig. 5h).

We will rephrase the interpretation of this spectrum. No we don't think the younger age has any meaning, and we attempt to clarify this in the paragraph on line 348.

Fig. 6. The whole explanation is insufficient and not comprehensible. What, for example, is meant by 'Model age spectrum of a simple age spectrum with two component system, mixing between more and less retentive argon gas reservoir.'?

The reviewer would best find answer to this in the original paper from where we reference this model age spectra. It is a model in that it is not real data, it is a hypothetical situation where an age spectra is obtained from degassing a two component system with a highly retentive domain and a less retentive domain in a step heating experiment. The original paper is Forster and Lister (2004).

Is the reviewer referring to the figure caption in this statement of it being 'insufficient and not comprehensible' or to the discussion in paragraphs on 238? If the former, we will try to flesh out the figure caption, however, aside from the caption we have provided, we are unsure what is being asked of us in this comment.